# A Comprehensive Evaluation of Code Language Models for Security Patch Detection

## Abstract

Detecting vulnerability-fixing commits (VFCs) is critical for timely security patch deployment, yet advisory databases lag patch releases by a median of 25 days and many fixes never receive advisories, driving interest in automated detection methods. We present a comprehensive evaluation of code language model (code LM) based VFC detection through a unified framework consolidating 20 fragmented datasets spanning more than 180.000 commits. Our analysis strenghtens existing observations through a systematic evaluation and reveals that high performance metrics mask fundamental limitations. Models achieve F1 scores of 0.9 when using full commits with messages, but drop to 0.6 on code alone while message-only models maintain close to original performance. This demonstrates reliance on textual patterns rather than semantic code understanding. We evaluate code LMs ranging from 125M to 30B parameters across multiple architectures, finding only marginal improvements with scale. Estimating out-of-distribution performance using repository-based splits exposes 10-11% performance drops compared to temporal splits, revealing models learn project-specific patterns rather than security semantics. Even additional intra-procedural context fails to improve detection. Prompt-based classification with models up to 480B parameters also underperforms fine-tuned approaches, indicating limitations beyond model scale. High inter-model agreement rates indicate convergence on similar patterns rather than complementary understanding. Overall, our findings suggest that code LMs appear fundamentally limited for code-centric security patch detection. We release our unified framework and evaluation suite to enable future SPD research.

## 1 Introduction

Modern software supply chains create cascading security dependencies where vulnerabilities impact numerous downstream projects. Organizations must identify and apply patches before attackers exploit the disclosure window. However, Imtiaz et al. (2023) reveal that advisory publication lags patch release by a median of 25 days, leaving systems vulnerable to attacks when developers rely on vulnerability databases such as the National Vulnerability Database (NVD) (NIST). Silent security patches present an even greater challenge, as they never receive public advisories, leaving downstream software vulnerable. As the vast majority of reused software is maintained in open-source repositories, an accurate analysis of each commit across upstream dependencies could result in safer software ecosystems with faster patch deployment times. However, as manual analysis is infeasible at scale, automated techniques for identifying vulnerability-fixing commits (VFCs) have become essential. Code LMs (Feng et al., 2020; Guo et al., 2022; Wang et al., 2021c; Li et al., 2023) have shown promise across code understanding tasks, leading to their adoption for Security Patch Detection (SPD), the task of identifying commits that fix vulnerabilities. While existing approaches report strong results when combining code with commit messages (Tang et al., 2023; Wang et al., 2021b; Tang et al., 2025; Sabetta & Bezzi, 2018; Nguyen-Truong et al., 2022; Chen et al., 2024; Sun et al., 2023a), the actual capabilities of these models for understanding security-relevant code changes without relying on textual information is largely unclear. Chen et al. (2024) and Sun et al. (2023a) performed ablation studies evaluating their systems with and without the commit message and observed a relevant drop in performance. Their findings suggest that models rely on both textural and code signals for effective prediction. Wang et al. (2023) proposed the first graph-based approach

that relies exclusively on code information, but requires expensive precomputations with limited demonstrated success (Wang et al., 2023; Wen et al., 2024).

We conduct the first comprehensive evaluation of transformer-based SPD, systematically examining models from 125M to 15.5B parameters across multiple architectures, context representations, and training strategies. Our experiments reveal that apparent success masks multiple fundamental limitations. Repository-based evaluation splits expose 10-11% performance drops compared to commonly-used temporal splits, indicating models memorize repository-specific patterns rather than learning transferable representations. This is in line with findings from Steenhoek et al. (2023) who report a similar performance drop for cross-project evaluations in VD on several systems. Scaling from CodeBERT (Feng et al., 2020) (125M) to Qwen3-Coder (Yang et al., 2025a) (30B) yields no meaningful improvement, suggesting architectural rather than capacity limitations. Furthermore, semantically-informed intra-procedural context enrichment evidently does not improve performance. Generally, models trained on `diffs` achieve roughly 0.6 F1 while message-only models reach 0.88, highlighting the reliance on textual shortcuts. Evaluation on commits with verified CWE mappings reveals near-random performance, indicating patterns learned from automatically-labeled samples fail to generalize to realistic security patches.

To facilitate training of commit-based detection approaches, a large corpus of security-annotated commits is required. This presents a key challenge that has primarily been addressed by matching commits from open source repositories to security advisories (Wang et al., 2021a; Lu et al., 2025) or identifying security patches based on a textual analysis of the commit message. As a variety of research groups have worked on this problem over the past decade with different target languages, labelling techniques and ground truths, a variety of fragmented datasets exist that make a large scale and comparative evaluation challenging. To address this, we build a comprehensive framework that unifies access to 20 VFC datasets, enabling systematic and comparative evaluation across different labeling strategies, programming languages, and vulnerability classes. Beyond addressing dataset fragmentation, we introduce a lightweight intra-procedural context enrichment method that efficiently identifies semantically relevant code context around modifications. Operating up to 33× faster than comparable graph-based techniques (Wang et al., 2023), our method provides additional statements based on data and control-flow analyses. However, even with semantically-informed context enrichment, model performance remains fundamentally limited, suggesting that current architectures fail to effectively learn suitable representations of security patches. The implications extend beyond SPD to modern development practices, where code review tools and LLM-assisted programming systems increasingly present changes in diff format. Understanding current reasoning capabilities on software changes is therefore critical for both security and general software engineering applications.

In summary, we make the following contributions: First, we provide a comprehensive evaluation revealing that current transformer-based SPD approaches fail across multiple dimensions including scale, context enrichment, and cross-repository generalization, suggesting fundamental limitations. Second, we develop a unified framework consolidating 20 fragmented datasets, enabling systematic comparative evaluation across different labeling strategies and programming languages. Third, we introduce a lightweight context enrichment method that efficiently provides semantically relevant context.

## 2 RELATED WORK

### 2.1 VFC DATASETS

Security research focusing on SPD or vulnerability detection (VD) requires code samples that are annotated with accurate vulnerability information. To approximate the underlying structural distribution of the data, a sufficiently large sample size is required. Yet, obtaining positive ground truth samples is difficult. Over the past decade an increasingly popular method of obtaining security annotated samples revolves around VFCs. To obtain code samples, open source repositories hosted on public version control systems are the primary source. The process of identifying which commit is security relevant has seen three major approaches: algorithmic pattern matching, often on the commit message (Zhou et al., 2019; Reis & Abreu, 2017; 2021; Zhou et al., 2022; Wartschinski et al., 2022; Sun et al., 2023b); linking vulnerability information from public advisories, bug trackers or similar sources to commits (Ponta et al., 2019; Ding et al., 2024; Bhandari et al., 2021; Nikitopoulos et al.,

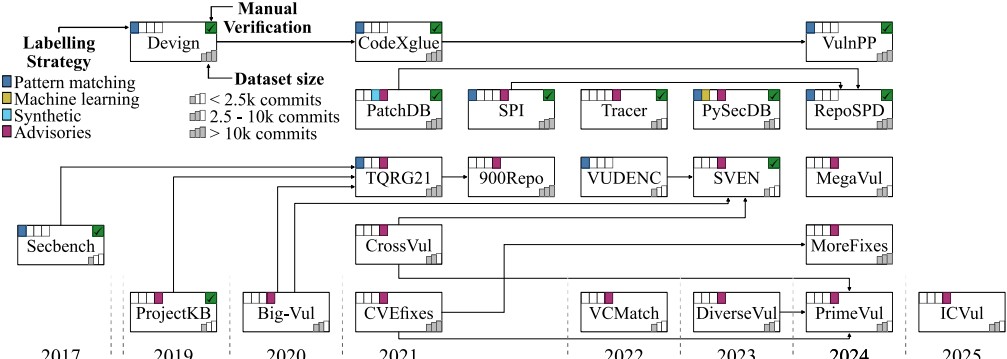

Figure 1: Temporal overview of existing VFC datasets, their label source, size and dependent datasets. The advisory label class includes other, similar, sources of information such as bug trackers. The synthetic label indicates that at least a subset of samples is synthetic. Manual verification indicates that some effort was spent validating at least a subset of the samples. VulnPP: VulnPatchPairs.

2021; Reis & Abreu, 2021; Wang et al., 2021a; Zhou et al., 2022; Lee & Chieu, 2021; Xu et al., 2022; Sun et al., 2023b; Ni et al., 2024; Akhoundali et al., 2024; Ding et al., 2025; Lu et al., 2025); and using other tools, algorithms or machine learning to perform code-based identification of potential VFCs (Wang et al., 2021a; Sun et al., 2023b; Risse & Böhme, 2024; Zhou et al., 2022; He & Vechev, 2023; Ponta et al., 2019). Additionally, synthetic generation of VFCs has also been explored (Wang et al., 2021a). Despite the technique used for generating labels, it has been shown (Ding et al., 2025) that the quality of labels is a significant concern. Some works have attempted to improve this by using manual verification (Reis & Abreu, 2017; Zhou et al., 2019; Ponta et al., 2019; Zhou et al., 2022; Xu et al., 2022; Sun et al., 2023b). An overview of dataset relations and key characteristics is shown in Figure 1. While some connections are present, overall several fragmented datasets exist, hindering large scale comparative evaluations. An in-depth summary of the presented datasets, their size, label distribution and included programming languages can be found in Table 7.

## 2.2 VFC DETECTION

Detecting VFCs has been explored under several assumptions, from SPD where the goal is to identify if a commit is security relevant (Wang et al., 2023; Wen et al., 2024; Zhou et al., 2022; Chen et al., 2024; Zhou et al., 2021; Yang et al., 2025b; Sun et al., 2023a), to ranking-based approaches trying to identify the VFC belonging to a specific advisory (Dunlap et al., 2024; Wang et al., 2022). For ranking-based approaches, a submodule is sometimes utilized that provides an SPD prediction indicator (Dunlap et al., 2024) but that is then embedded into a larger prediction framework that relies on additional metadata that is generally used for ranking VFCs. For SPD several works have also explored the prediction capabilities under the utilization of the code changes along with the commit message (Tang et al., 2023; Wang et al., 2021b; Tang et al., 2025; Sabetta & Bezzi, 2018; Nguyen-Truong et al., 2022). In this work we evaluate SPD under the limitation that only the code is utilized for the classification to understand the security-related code reasoning capabilities. Sequence-based approaches typically rely on representing code changes as token sequences that are used for training a machine learning model or ensemble of models for supervised classication (Sun et al., 2023a; Dunlap et al., 2024; Zhou et al., 2021). Zhou et al. (2023a) explore a contrastive learning approach to pretrain a BERT model with a specialized embedding space that is used in downstream SPD systems. Wang et al. (2023) introduced PatchCPG, a novel semantics-aware patch representation based on code property graphs (Yamaguchi et al., 2014a). To generate a PatchCPG, the CPG is generated for the pre- and post-patch versions using Joern (Yamaguchi et al., 2014b). They then perform dependence-guided forward/ backward slicing, seeded by the added/ deleted lines, to confine context. Finally, it feeds a multi-attributed graph into a graph neural network (GNN) to classify security patches directly from graph structure (Wang et al., 2023). Building on this idea, Wen et al. (2024) construct a RepoCPG that preserves semantic changes while augmenting them with cross-file dependencies (e.g., function-level call relations), and fuses graph-based and sequence-based representations with progressive learning to capture relationships among multiple code changes at repository scale (Wen et al., 2024). Recently, a dynamic approach to evaluating

| DS | PLs | Label | Commits | | Projects | |
|----|-----|-------|---------|---------|---------|-------|
| | | | Total | VFCs | 100% | 75% |
| $\mathcal{D}_1$ | C/C++ | MR | 55 789 | 36% | 462 | 7 |
| $\mathcal{D}_2$ | C/C++ | Adv | 82 566 | 26% | 852 | 29 |
| $\mathcal{D}_3$ | C/C++ | All | 89 963 | 33% | 1355 | 40 |
| $\mathcal{D}_4$ | Multi | All | 180 348 | 29% | 7587 | 309 |

(a) Dataset characteristics

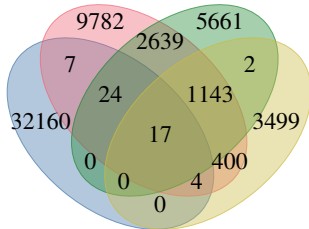

(b) Source overlap in $\mathcal{D}_1$.

Figure 2: Evaluation dataset compositions (a) and source set overlap in $\mathcal{D}_1$ from the largest four contributors: PatchDB (Wang et al., 2021a) (blue), SPI-DB (Zhou et al., 2022) (pink), RepoSPD (Wen et al., 2024) (Green), Devign (Zhou et al., 2019) (yellow)

safe patches by integrating dynamic symbolic execution has also been proposed (Luo et al., 2024). Additionally, Yang et al. (2025b) propose an LLM-based generative approach that utilizes ressource augmented generation (RAG) through embedding based matching of historic VFC information to detect VFCs.

# 3 VFC COLLECTION FRAMEWORK

Machine learning performance depends on both data quality and quantity. The fragmented landscape of VFC datasets (Section 2.1) presents challenges for comparative evaluation, with accessibility issues and format inconsistencies hindering reproducibility.

To address these challenges, we develop a comprehensive framework that unifies access to existing VFC datasets through systematic parsing, normalization, and enrichment. The framework consists of three stages: data ingestion and normalization across diverse formats and platforms, enrichment with commit metadata and repository information, and deduplication and filtering for dataset customization. The framework currently integrates 20 different data sources, enabling researchers to combine and filter collections based on specific research requirements. To manage the overlap during dataset merging, we implement three complementary deduplication strategies. First, identical entries (same commit hash and repository) are merged to maximize metadata retention while removing conflicts where labels disagree ($\mathcal{D}_1$: 998, $\mathcal{D}_2$: 9441, $\mathcal{D}_3$: 5706, $\mathcal{D}_4$: 5706). Second, semantic matches based on diff content and modified files are removed, primarily catching commits across repository mirrors. Third, we introduce latent-space deduplication using UniXcoder embeddings with cosine similarity to reduce semantic redundancy while preserving dataset diversity. When developing systems on these datasets, especially when considering commit messages, care must be taken to avoid learning the patterns that were used to acquire and label the samples. To attempt to identify spurious correlations within the `diffs`, we performed a comprehensive effect size analysis (Section A.1). While VFCs exhibit statistically significant differences in structural features, such as fewer code modifications and increased memory operations, all effect sizes remain negligible. The framework offers extensive filtering capabilities, allowing researchers to create custom datasets based on, among others, different programming languages, vulnerability classes, labeling strategies, source datasets, and temporal constraints.

Using our framework, we systematically construct four datasets with increasing scope to evaluate the impact of data quality, quantity, and diversity on SPD detection performance. An overview of the size and label distribution for each dataset is shown in Table 2a. A detailed breakdown of all contributing source datasets to the respective target datasets can be seen in Table 7 and the source set overlap of the four largest contributors in $\mathcal{D}_1$ is shown in Figure 2b.

$\mathcal{D}_1$ **- Manually reviewed, Advisory-based, C/ C++** : Our core dataset combines all C/ C++ commits across all vulnerability classes from datasets where the original authors performed some manual validation of the label quality. This dataset is used to establish performance baselines and as a target for cross-dataset evaluations. However, label inaccuracy presents an open and unquantified issue throughout this and the following datasets.

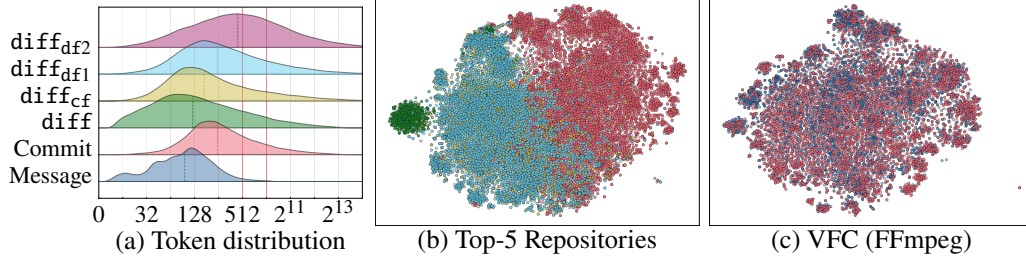

|       |       |       |
|:-----:|:-----:|:-----:|
| (a) Token distribution | (b) Top-5 Repositories | (c) VFC (FFmpeg) |

Figure 3: Dataset characterization for $\mathcal{D}_1$. (a) Token size distributions (log-2 scale) for different representations: commit messages alone (Message), code changes alone (Diff), complete commits (Commit), and enriched diffs with varying context levels. Red lines indicate common transformer sequence limits. Tokens are generated using CodeBERT tokenizer. (b) t-SNE visualization of code changes from the 5 most common repositories based on UniXcoder embeddings: FFmpeg (50.3%, orange), Linux (24.1%, light blue), media-tree (19.5%, yellow), OpenBSD (4.4%, green), Ceph (1.7%, dark blue). (c) t-SNE visualization of FFmpeg commits colored by VFC labels: VFCs (43.4%, red), Non-VFCs (56.6%, blue).

$\mathcal{D}_2$ **- Advisory-based, `C/ C++`** : Extended version of $\mathcal{D}_1$ that removes the restriction on additional systematic manual verification.

$\mathcal{D}_3$ **- Comprehensive, `C/ C++`** : All `C/ C++` commits, including, among others, those labeled using machine-learning approaches and traditional tools.

$\mathcal{D}_4$ **- All commits**: This is the largest possible dataset removing all filtering constraints including the target language to see if cross-training on this comprehensive dataset and evaluating on the baseline improves model performance.

To understand $\mathcal{D}_1$ characteristics, we analyze token distributions and latent representations (Figure 3). Commit messages fit within 512 tokens, while 23.3% of full commits and 16.4% of `diff`s exceed this limit (see Table 8 for details). A t-SNE analysis of UniXcoder (Guo et al., 2022) embeddings reveals clear repository-specific clusters, motivating repository-based splits for realistic generalization assessment. Within repositories, VFC and non-VFC commits largely overlap, suggesting vulnerability patterns are not trivially distinguishable.

## 4 CONTEXT ENRICHMENT

Code sequential SPD approaches rely on syntactic changes from version control systems. However, these changes are very precise and may lack sufficient semantic context for accurate classification. Consider the memory leak fix in Listing 1. The actual changes (yellow) specify what was modified, but without additional context (green), the impact remains unclear. To address this limitation, we propose a lightweight data-flow-based context enrichment method. Our approach preserves relevant intra-procedural program context around modified statements without including entire functions. Algorithm 1 formalizes this process. Our method begins by identifying changed files using `git diff`. We then extract functions[1] that intersect with the diff hunks. For each changed function, we generate concrete syntax trees (CSTs) for both pre- and post-patch versions using tree-sitter (Brunsfeld). We then build a statement-level intermediate representation (IR) from the CST for each version. Using GumTree (Falleri & Martinez, 2024), we create a structural diff that maps CST nodes to edit categories (`ADD, DEL, MOD`). From the changed nodes, we identify relevant data-flow statements through lightweight forward and backward slicing along definition-use chains. We perform slicing at depths $d \in \{1, 2\}$ to capture varying context levels. Relevant statements are marked as context in the IR and merged into a single function-level representation. We then augment this with minimal control-flow context by including immediate enclosing structures (if-statements, loops, etc.). The final representation uses diff-style formatting with explicit change tokens. The green statements in Listing 1 demonstrate our context enrichment. With this additional context, we can see that the pointer is properly freed in the fail case, information not apparent from the changed lines alone. To

---

[1]The current implementation targets `C` functions but can be extended to other languages.

**Algorithm 1:** Context enriched `diff`

**Input:** Pre-patch $c_p$ and target commit $c_t$.
**Output:** Enriched diff $\mathcal{D}_{ctx}$
$\mathcal{D} \leftarrow \text{diff}(c_p, c_t)$
$\mathcal{F} \leftarrow \text{changed\_functions}(\mathcal{D})$
**for** *each $f$ in $\mathcal{F}$* **do**
   // Concrete syntax tree
   $CST_p^f \leftarrow \text{TreeSitter}(c_p, f)$
   $CST_t^f \leftarrow \text{TreeSitter}(c_t, f)$
   // Statement level IR
   $(IR_p^f, IR_t^f) \leftarrow \text{build\_ir}(CST_p^f, CST_t^f)$
   // Compute structural diff
   $\Delta^f \leftarrow \text{GumTree}(CST_p^f, CST_t^f)$
   // Changed statement extraction
   $(S_\Delta^p, S_\Delta^t) \leftarrow \text{extract\_statements}(\Delta^f)$
   **for** *each $k$ in $\{p, t\}$* **do**
      // Bi-directional slicing
      $S_{ctx}^{bw} \leftarrow \text{backward\_slice}(S_\Delta^k, CST_k^f, d)$
      $S_{ctx}^{fw} \leftarrow \text{forward\_slice}(S_\Delta^k, CST_k^f, d)$
      $IR_k^f \leftarrow \text{update\_ctx}(IR_k^f, S_{ctx}^{bw}, S_{ctx}^{fw})$
   **end**
   // Merge and align IRs
   $IR^f \leftarrow \text{merge\_ir}(IR_p^f, IR_t^f, \Delta^f)$
   $IR^f \leftarrow \text{add\_control\_flow}(S_\Delta^p, S_\Delta^t)$
   $\mathcal{D}_{ctx} \leftarrow \text{append}(\mathcal{D}_{ctx}, IR^f)$
**end**
**return** $\mathcal{D}_{ctx}$

```
Commit: a863c97e
Fix memory leak in
    smoothstreamingenc.c
```

```
static void get_private_data(OutputStream *os)
{
  AVCodecContext *codec =
      os->ctx->streams[0]->codec;
  uint8_t *ptr = codec->extradata;
  int size = codec->extradata_size;
  int i;
  if (codec->codec_id == AV_CODEC_ID_H264) {
    ff_avc_write_annexb_extradata(ptr, &ptr,
     &size);
    if (!ptr)
      ptr = codec->extradata;
  }
  if (!ptr)
    return;
  os->private_str = av_mallocz(2*size + 1);
  if (!os->private_str)
  [MOD] [- return; -] {+ goto fail; +}
  for (i = 0; i < size; i++)
    snprintf(&os->private_str[2*i], 3, "%02x",
      ptr[i]);
[ADD] fail:
  if (ptr != codec->extradata)
    av_free(ptr);
}
```

Listing 1: Changed function from ffmpeg commit a863c97e, diff lines in yellow with word-level highlighting and GumTree line annotation. Additional context lines based on data-flow and enclosing control flow in green.

reduce diff size, we track changes at word-level granularity within lines, explicitly showing removed and added words rather than treating modified lines as separate deletions and additions.

Our approach significantly outperforms Joern-based (Yamaguchi et al., 2014b) analysis used by GraphSPD (Wang et al., 2023). On 500 PatchDB (Wang et al., 2021a) samples, our method averaged 1.72 seconds per sample versus 57.36 seconds for GraphSPD. Figure 3(a) shows token size distributions of all context levels for $\mathcal{D}_1$. We generate standard diffs following Dunlap et al. (2024) but remove comments (see Section A.5). Our three enrichment levels show progressive size increases: control-flow structures ($\text{Diff}_{cf}$) increase median size from 122 to 170[2] tokens (1.4×), immediate data-flow dependencies ($\text{Diff}_{df1}$) to 247 tokens (1.45×), and a second data-flow pass ($\text{Diff}_{df2}$) to 450 tokens (1.8×). The impact of context enrichment on VFC detection accuracy is evaluated in Section 5.2.

# 5 EVALUATION

To better understand the capabilities of current code LMs for SPD under realistic conditions and to evaluate the benefit of additional intra-procedural context, we conduct a comprehensive evaluation across multiple datasets, models, and data splitting strategies.

## 5.1 MODELS AND TRAINING SETUP

We evaluate popular code LMs across different sizes, pre-training tasks and architectures (Table 1). Following Ding et al. (2025), we use 80/10/10 splits and select the best validation F1 checkpoint

---

[2] Token counts use the CodeBERT tokenizer but are similar across tokenizers.

Table 1: Overview of evaluated models with architectural details and specifications.

| Model | Param | Tokens | Arch | Pre-training |
|---|---|---|---|---|
| CodeBERT (Feng et al., 2020) | 125M | 512 | Enc | PL + NL |
| UniXcoder (Guo et al., 2022) | 125M | 512/ 1024 | Enc-Dec | PL + NL + AST |
| CodeBERT C++ (Zhou et al., 2023b) | 125M | 512 | Enc | C++ Code |
| CodeT5 Large Wang et al. (2021c) | 770M | 512 | Enc-Dec | PL + NL |
| CommitBART (Liu et al., 2022) | 140M | 1024 | Enc-Dec | Commits |
| StarCoder Li et al. (2023) | 15.5B | 2048[1] | Dec | PL |
| Qwen3-Coder Yang et al. (2025a) | 30B | 4096[1] | Dec | PL + PL |

[1] The models max capacity is larger.
PL: Programming Language, NL: Natural Language, AST: Abstract Syntax Tree.

for testing. All experiments use an AMD EPYC 9534 CPU with 2.95TB RAM and four NVIDIA H200 NVL GPUs. Models with less than 1B parameters are trained for 7 epochs (batch size 64, AdamW (Loshchilov & Hutter, 2019), lr=$2 \times 10^{-5}$) with a single layer linear classification head. StarCoder is trained with a batch size of 4 with two gradient accumulation steps. To train Gwen Yang et al. (2025a) we use low rank adaptation (Hu et al., 2022) with rank 16 and a batch size of 24, 2 gradient accumulation steps and train for 4 epochs. Class imbalance is handled via inverse frequency weighting. For context-enriched representations, change identifiers are added as special tokens with embeddings initialized from related existing tokens. We report F1-scores as a well recognized metric and adapt the vulnerability detection score (VD-S) introduced by Ding et al. (2025) as the patch detection score PD-S (= VD-S) = $FNR@(FPR \leq r)$ where $r \in [0, 1]$ (Ding et al., 2025) to quantify detection rates under constrained misclassification rates to help mitigate base rate fallacy (Arp et al., 2022). We set $r = 0.05$ as suggested by the original authors. Model inference times and memory usage for all fine-tuned models are reported in Section A.3.

## 5.2 Experimental Results

To understand the current capabilities we first determine the impact the commit message and `diff` have on the performance respectively. Then, following the findings from Section 3 regarding repository structure in the models latent space we investigate different splitting strategies. Using a repository-based split we then determine the model impact across several context levels to determine if larger models with additional context lead to a performance increase. Additionally we investigate if training on larger but noisier samples improves the model performance but find no significant difference (See Section A.7). To assess training stability, we conduct experiments with three random seeds and repository splits, finding very stable training results with standard deviations below 1.5% for PD-S and F1 scores (see Section A.6).

**Commit Message Impact** Following existing observations (Chen et al., 2024; Sun et al., 2023a) we quantify the relative contributions of the commit message (NL) and `diff` (PL) for SPD by conducting ablation studies with two models. We train CodeBERT (Feng et al., 2020) and CodeT5-Large (Wang et al., 2021c) on $\mathcal{D}_1$ under random split conditions to establish baseline performance characteristics (see Table 2). Models trained on complete commits achieve F1-scores of 0.89 for CodeBERT and 0.9 for CodeT5-Large respectively while the PD-S score is relatively low at 0.31 and 0.32. While these scores indicate usable models, when removing the commit message, the F1 score decreases by 30%. Conversely, when only training on the commit message, the performance is within 2% of the baseline. This clearly demonstrates the reliance on the commit message for classification when the full commit is utilized and masking the models code reasoning capabilities. The studies of Sun et al. (2023a) and Chen et al. (2024) also support this observation. Chen et al. report a significant reduction in F1 score by 22% when commit messages are dropped. Sun et al. only report AUC values, their utility is most influenced by the removal of commit messages, while code-only approaches score worst. Unlike our result, their numbers suggest a higher increase in utility when the combination of diff and message is considered. Detailed learning dynamics for our experiments can be seen in Section A.7 that underline the training stability. While not directly comparable due to smaller/ different training sets, the performance achieved on full commit classification is generally in line with performances reported by Zhou et al. (2022) and higher than other systems (Tang et al., 2025; 2023; Chen et al., 2024) that also utilize the commit message but are trained on individual

Table 2: SPD performances for different data splitting strategies and input contexts. Detailed training dynamics are shown in Section A.7.

| Model | Context | Random | | Temporal | | Repository | | CWE | |
|---|---|---|---|---|---|---|---|---|---|
| | | F1 ↑ | PD-S ↓ | F1 ↑ | PD-S ↓ | F1 ↑ | PD-S ↓ | F1 ↑ | PD-S ↓ |
| CB | Commit | 0.89 | 0.31 | 0.89 | 0.37 | 0.79 | 0.45 | 0.82 | 0.46 |
| | Message | 0.87 | 0.37 | 0.88 | 0.37 | 0.77 | 0.47 | 0.82 | 0.42 |
| | $\texttt{diff}_{\texttt{git}}$ | 0.61 | 0.90 | 0.68 | 0.89 | 0.59 | 0.81 | 0.54 | 0.97 |
| CT5L | Commit | 0.90 | 0.32 | 0.90 | 0.47 | 0.81 | 0.45 | 0.81 | 0.49 |
| | Message | 0.90 | 0.42 | 0.89 | 0.45 | 0.79 | 0.45 | 0.82 | 0.45 |
| | $\texttt{diff}_{\texttt{git}}$ | 0.63 | 0.88 | 0.69 | 0.87 | 0.61 | 0.83 | 0.52 | 0.98 |

CB: CodeBERT, CT5L: CodeT5 Large.

[1] The CWE split is balanced

datasets. In general, this dependency on textual signals, while effective for practical deployment scenarios where messages are available, raises fundamental questions about the capability of current code LMs to identify vulnerability-fixing patterns from code changes alone.

**Splitting Strategy Impact** Data leakage and spurious correlations are well-documented challenges faced in machine learning for security (Arp et al., 2022) that can cause severe overestimation of performances. Additionally, a shift of the underlying data over time has to be considered under realistic evaluation of detection systems. (Ding et al., 2025; Kan et al., 2024). To quantify the robustness of a representation and model against this, a temporal split is generally considered. However, we argue that for the data at hand we are additionally able to estimate out distribution generalization capabilities by splitting across repositories. This is also supported by Steenhoek et al. (2023) who have identified an average decrease in F1 score of 0.11 for various systems on VD when using cross-project evaluations. Figure 3 attempts to visualize inherent structural biases across different repositories by running a t-SNE analysis on the hidden latents of the samples of the five largest repositories in $\mathcal{D}_1$. Results for random, temporal and repository based splits are shown in Table 2. Interestingly, there seems to be no degradation by temporal splitting (within the evaluated timeframe), but the generalizability seems limited indicated by an F1 decrease of 11% across both models under cross-project evaluations. To compare results to graph-based approaches, we evaluate GraphSPD Wang et al. (2023) under repository split conditions. We trained the model on the dataset used by the authors (PatchDB (Wang et al., 2021a))[3] and evaluated the model using 500 test samples from $\mathcal{D}_1$ that were in the repository distribution seen during training and 500 that were not. This caused an F1 performance decrease of 30% from 0.46 to 0.32. For Grape we were unable to recreate results due to a missing implementation (Han et al., 2024).

These results suggests that models rapidly exhaust the learnable signal in the data and subsequently engage in memorization of training-specific patterns that fail to generalize across repository boundaries. To ensure robust evaluation we adopt the repository split strategy for all subsequent experiments. While it would be ideal to additionally ensure temporal separation, the reduction in sample size would be too severe.

**CWE Impact** As the data quality is a key factor for model performance and even in $\mathcal{D}_1$ the majority of commits is labelled as VFC based on some rulset or other approximation the label noise is a significant concern. Additionally, the question arises how well these autmatically labelled samples approximate the distribution of realistic security patches. To investigate this empirically we create an additional split where all VFCs that are mapped to a CWE are placed in the test set. This contributes 2500 samples that are labeleld vulnerable. To balance the test set we iterate over each VFC and draw the closest benign sample based on the distance in the embedding space of a non-finetuned UniXcoder (see Section 2.1). This creates a balanced test set that contains 10% of samples from $\mathcal{D}_1$. While it would be ideal to additinally ensure repository separation between the different splits, this would result in too small splits. Results clearly indicate that especially for $\texttt{diff}$-only classification the performance degrades significantly with F1 scores close to 0.5 and PD-S scores close to 1. This

---

[3]We did not evaluate on the full $\mathcal{D}_1$ set due to already very long preprocessing times.

Table 3: Code-centric SPD performances across different model classes, sizes and pre-training tasks.

| Metric | Context | CB | CB++ | UC$_{512}$ | UC$_{1024}$ | CT5L | CommitBart | SC | Gwen3-C |
|---|---|---|---|---|---|---|---|---|---|
| **F1 ↑** | diff$_{\text{git}}$ | 0.59 | 0.59 | 0.59 | 0.59 | 0.61 | 0.53 | 0.63 | **0.66** |
| | diff$_{\text{word}}$ | 0.60 | 0.61 | **0.61** | **0.61** | 0.59 | **0.58** | 0.60 | 0.62 |
| | cf | **0.61** | **0.62** | **0.61** | 0.59 | **0.62** | 0.57 | 0.60 | 0.63 |
| | df$_1$ | 0.58 | 0.59 | 0.59 | 0.59 | 0.59 | 0.55 | 0.62 | 0.61 |
| | df$_2$ | 0.57 | 0.58 | 0.59 | 0.58 | 0.58 | 0.55 | **0.64** | 0.61 |
| **PD-S ↓** | diff$_{\text{git}}$ | 0.81 | 0.81 | **0.80** | **0.79** | 0.83 | **0.80** | 0.82 | 0.79 |
| | diff$_{\text{word}}$ | 0.81 | **0.80** | **0.80** | 0.80 | 0.81 | 0.84 | **0.79** | **0.78** |
| | cf | **0.78** | **0.80** | **0.80** | **0.79** | **0.79** | 0.83 | 0.82 | 0.82 |
| | df$_1$ | 0.82 | 0.82 | 0.82 | 0.81 | 0.82 | 0.82 | 0.83 | 0.86 |
| | df$_2$ | 0.85 | 0.85 | 0.81 | 0.83 | 0.81 | 0.89 | 0.82 | 0.83 |

CB(++): CodeBERT (C++), UC: UniXcoder, CT5L: CodeT5 Large, SC: StarCoder, Qwen3-C: Qwen3-Coder. cf: diff$_{\text{word}}$ + control flow enclosure, df$_i$: cf + $i$ data-flow pass(es)

indicates that the patterns learned on the remaining samples do not generalize well to the realistic CWE-labelled samples.

**Model Impact** To investigate how model architecture, size, context length and specialized pretraining affect SPD performance we evaluate all models shown in Table 1 on the diff representation. We establish baselines using standard CodeBERT (Feng et al., 2020) and UniXcoder (Guo et al., 2022) models (both 125M parameters) and include CodeBERT C++ (Zhou et al., 2023b) and CommitBART (Liu et al., 2022) to assess specialized pre-training impact. As shown in Table 3, baseline models achieve F1-scores around 0.6 with the specialized models showing no advantage. Extended context capacity (UniXcoder with 1024 tokens) yields no improvement despite many diffs exceeding 512 tokens (see Figure 3). Scaling to CodeT5 Large (Wang et al., 2021c), StarCoder2 (Li et al., 2023) and Gwen3-Coder (Yang et al., 2025a), also only produces marginal gains suggesting diff-based SPD faces fundamental limitations rather than model capacity constraints on current dataset sizes.

**Context Enrichment** Incrementally increasing the intra-procedural context by control-flow enclosures (cf), a single data-flow pass (df$_1$) and two data-flow passes (df$_2$) across the previously evaluated models interestingly does not cause incremental performance increases (see Table 3). While a small performance increase is especially visible in the PD-S when enriching diffs with the immediate control-flow enclosures, adding additional data-flow context seems to cause slight performance decreases across model classes. These results suggest that rather than providing relevant semantic information, the additional data-flow context seems to contribute more to noise, indicating that intra-procedural context may not be sufficient to capture the required semantics.

**Failure Case Analysis** To understand the interactions between the samples (representations) and models we investigate model agreement rates across various model combinations on the same train-test set. Interestingly, the mean agreement rate is universally high (see Section A.2 for a detailed breakdown). For all eight models evaluated on the diff representation, the mean agreement rate is 87% with all models predicting unanimously in 48% of cases. When comparing agreement rates of the same model trained using three different seeds on the same splits, mean agreement rates of 94% can be observed. This strongly suggests that the emerging patterns learned during training align closely across different training runs and even show similarity across diverse models. For the enriched representations, similar mean agreement rates between 88% and 90% can be observed. When comparing the agreement rates across classes, a shift can be observed, where the models trained with additional context agree more on benign samples whereas they agree slightly less for vulnerable samples. Following this, within these agreement classes, the enriched models have higher correctness rates on benign samples and lower rates for vulnerable samples compared to the baseline models trained on diffs. These convergent behaviors across diverse architectures and training conditions suggest that current transformer-based approaches may be reaching a fundamental ceiling in their ability to distinguish security-relevant code changes from benign modifications when relying solely on code changes. Manual inspection of the top 50 unanimous failure cases reveals that false positives are commonly small changes that exhibit defensive hardening (47%) or memory operation (36%) patterns, i.e. changes addressing bugs that may be security-dependent but are

Table 4: F1 scores for prompt-based SPD using Qwen3-Coder 480B across different context levels. CoT: Chain-of-Thought prompting, Logit: constrained vocabulary with logit-based classification.

| Diff Format | diff | | cf | | df$_1$ | | df$_2$ | |
|---|---|---|---|---|---|---|---|---|
| | CoT | Logit | CoT | Logit | CoT | Logit | CoT | Logit |
| git | 0.40 | 0.33 | 0.43 | 0.36 | 0.43 | 0.40 | **0.47** | 0.43 |
| word | 0.38 | 0.32 | 0.41 | 0.33 | 0.43 | 0.35 | 0.45 | 0.38 |

indistinguishable without broader context. False negatives tend to involve larger multi-file changes or complex semantic modifications. These observations suggest that significant improvements may require either fundamentally different architectural approaches or access to substantially richer semantic representations (see Section A.2 for details).

**Prompt-based Classification** To compare the performances of fine-tuned models to prompt-based classification approaches for SPD, we evaluate Qwen3-Coder-480B (Yang et al., 2025a) with activation aware quantization (AWQ) (Lin et al., 2023)QuantTrio on the same test split from $\mathcal{D}_1$ used in previous experiments. Based on observations made by Sprague et al. (2025) that chain-of-thought prompting is not universally useful, we adapt two different prompting strategies to evaluate their effectiveness for SPD. We investigate chain-of-thought (CoT) reasoning and logit-based classification where the vocabulary is constrained to two tokens (`benign`, `VFC`) and classification is performed based on the probability distribution over these tokens. Additionally we evaluate the model on both the `git` and `word`-based formatting to see if this impacts performance. The CoT prompt template for `git`-based diffs is provided in Section A.8. We evaluate on 3675 samples from $\mathcal{D}_1$ under repository-based splitting across all context enrichment levels (Table 4). Overall, prompt-based approaches achieve substantially lower F1 scores than fine-tuned models, with the best CoT configuration reaching 0.47 on df$_2$ compared to 0.6+ for fine-tuned models on baseline `diffs`. CoT consistently outperforms logit-based classification, suggesting the model benefits from explicit reasoning steps. Context enrichment shows consistent improvements, with F1 scores increasing from 0.38/0.40 on `diff` to 0.45/0.47 on df$_2$, indicating that additional semantic context provides some relevant signal. However, this shows that despite the significantly larger model size, general purpose reasoning capabilities do not outperform smaller fine-tuned models for SPDwithout additional specialized training or capabilities. Iterative, tool-based explorations could offer potential avenues for future improvement.

## 6 CONCLUSION

We present a comprehensive evaluation of code LM-based SPD through a unified framework spanning 20 datasets. Our analysis strengthens existing findings and highlights common pitfalls that can can mask fundamental issues behind good performance metrics by exploiting textural shortcuts rather than understanding code semantics. When messages are removed, performance drops substantially while message-only models maintain performance, indicating a strong dependence on textual rather than structural patterns. Experiments reveal that neither model scaling nor semantically-informed intra-procedural context enrichment yields meaningful improvements, with all architectures converging on similar detectable samples. Repository-based evaluation exposes generalization failures, demonstrating the need for robust out-of-distribution evaluations. These findings suggest that either additional context or significantly larger distributions and models are required to learn robust and generalizable representations that achieve usable results. Our framework unifies access to existing datasets and enables future SPD research toward code-centric VFC detection.

**Data Availability.** To facilitate reproducibility and future research, we release our unified VFC framework at [anonymous submission], which provides standardized access to all 20 integrated datasets with consistent preprocessing and deduplication. The training pipeline is integrated into the system developed by Ding et al. (2025) at [anonymous submission]. Our lightweight context enrichment algorithm is published separately at [anonymous submission] with documentation for integration into existing workflows.

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

# A APPENDIX

The following appendix provides an in-depth overview of all evaluated VFC datasets (Section A.1) and an evaluation of model agreement rates (Section A.2). Additionally we provide an overview of model infrence times and memory consumption in Section A.3 as well as detailed token size statistic in Section A.4. Visualizations of the training dynamics for all experiments are shown in Section A.6. Additionally we empirically discuss the observed training stability (Section A.6) and provide additional experiments that evaluate cross-dataset training performances (Section A.7).

## A.1 VFC DATASET OVERVIEW AND CHARACTERISTICS

A detailed overview of evaluated VFC datasets from the past decade, and their contribution to datasets established in Section 3, can be seen in Table 7. To better understand and quantify the distinguishing

Table 5: Effect size analysis of VFC characteristics in $\mathcal{D}_1$. Cohen's d values indicate the standardized mean difference between VFC and non-VFC commits, with negative values indicating lower values for VFCs. All p-values are significant after FDR correction (*** p ¡ 0.001, * p ¡ 0.05).

| Feature | Cohen's d | VFC Mean | Non-VFC Mean | p-value | FDR p |
|---|---|---|---|---|---|
| *Code Change Metrics* | | | | | |
| Number of `[ADD]` | −0.084 | 9.02 | 16.29 | <0.001 *** | <0.001 |
| Number of `[DEL]` | −0.079 | 1.47 | 3.16 | <0.001 *** | <0.001 |
| Number of `[MOD]` | −0.124 | 2.69 | 5.03 | <0.001 *** | <0.001 |
| Token Count | −0.081 | 359.06 | 533.58 | <0.001 *** | <0.001 |
| | | | | | |
| *Semantic Patterns* | | | | | |
| Memory Operations (%) | 0.173 | 30.30 | 22.79 | <0.001 *** | <0.001 |
| Security Keywords (%) | −0.053 | 24.43 | 26.77 | <0.001 *** | <0.001 |
| Error Handling (%) | −0.108 | 45.01 | 50.40 | <0.001 *** | <0.001 |
| Concurrency Patterns (%) | −0.022 | 12.71 | 13.46 | <24 * | <0.024 |
| | | | | | |
| *Top tokens* | | | | | |
| `struct` | −0.152 | 0.23 | 0.84 | <0.001 *** | <0.001 |
| `static` | −0.105 | 0.39 | 0.72 | <0.001 *** | <0.001 |
| `else` | −0.091 | 0.29 | 0.50 | <0.001 *** | <0.001 |
| `return` | −0.084 | 0.93 | 1.33 | <0.001 *** | <0.001 |

characteristics between VFC and non-VFC commits in $\mathcal{D}_1$, we conducted a comprehensive effect size analysis using *Cohen's d*. Table 5 presents the results for various structural and semantic features extracted from $\mathcal{D}_1$. While the analysis reveals several patterns, all effect sizes fall within the negligible range ($|d| < 0.2$), indicating that while these differences are statistically significant, they represent subtle distinctions that do not highlight significant shortcuts in the training set. All reported p-values remain significant after false discovery rate (FDR) correction, confirming the robustness of these findings.

## A.2 MODEL AGREEMENT

As previously described the training with different models, on different representations, generally, leads to a similar performance plateau that reflects poor detection results. To try to gain insights into the reason for this plateau we evaluate model agreement rates on the same test set. Interestingly these agreement rates are much higher than anticipated considering the performance and indicate that the models learn similar representations or patterns resulting in very similar classifications (see Figure 4). These high agreement rates hold across different models for the same representation and even across representations, suggesting that the context does not lead to successful classification of different subsets of samples.

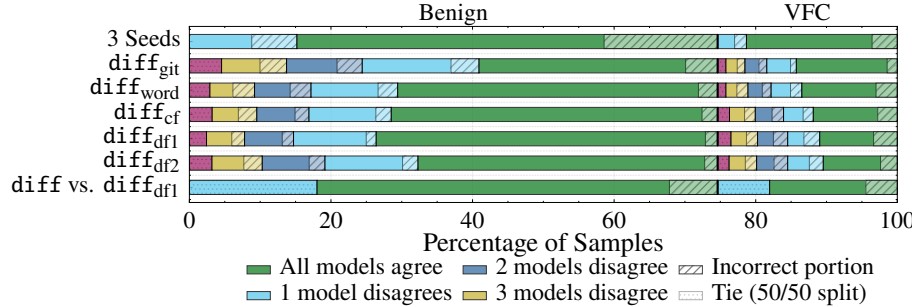

Figure 4: Model agreement analysis on test sets of enriched samples for different model combinations.

**Manual Failure Case Analysis.** To further investigate model behavior, we manually evaluated the top 50 failure cases where all 8 models agreed on the incorrect label based on the $\texttt{diff}_{word}$ representation. Samples were sorted based on the average model confidence. This resulted in 36 false positives and 14 false negatives. Upon manual review, we observed that these commits are generally small with 18 containing no deletions and 11 containing only a single line addition. The most common pattern observed was defensive hardening or error handling, present in 47% of false positive samples, followed by distinct memory operations in 36% of cases. These patterns emerged prominently due to the small size of the commits and are likely the reason for the high model confidence. These commits commonly address bugs or edge cases for which a case could be made that they may be security-dependent. Evaluating this without extensive additional semantic context is challenging, highlighting the need for richer representations. For false negative cases, no clear patterns could be observed; however, several samples were either relatively large multi-file changes or involved complex semantic changes that also require additional context to fully comprehend.

## A.3 MODEL INFERENCE BENCHMARKS

To provide insights into the computational requirements of different model architectures for SPD, we benchmark inference performance across all fine-tuned models. Table 6 presents detailed performance metrics including inference time, memory consumption, and throughput. All measurements were conducted on a single NVIDIA H200 NVL GPU using sequential inference (batch size of 1) over 1000 samples following an initial warmup

phase. Samples were generated for each model to evaluate under maximum load conditions.

Table 7: Overview of research-based data sets that contain VFCs. PM=Pattern Matching on commit messages, MR=Manual Review, A=Advisory, SM=Semantic Matching, SYN=Synthetic, BT=Bug Tracker, O=Other. Citations for each dataset follow: Secbench (Reis & Abreu, 2017), Project KB (Ponta et al., 2019), Devign (Zhou et al., 2019), Big-Vul (Ding et al., 2024), CVEfixes (Bhandari et al., 2021), SPI (Zhou et al., 2022), PatchDB (Wang et al., 2021a), TQRG21 (Reis & Abreu, 2021), CrossVul (Nikitopoulos et al., 2021), CodeXGLUE (Lu et al., 2021), 900Repo (Lee & Chieu, 2021), VCMatch (Wang et al., 2022), Tracer (Xu et al., 2022), Vudenc (Wartschinski et al., 2022), DiverseVul (Chen et al., 2023), PySecDB (Sun et al., 2023b), Sven (He & Vechev, 2023), MoreFixes (Akhoundali et al., 2024), MegaVul (Ni et al., 2024), VulnPatchPairs (Risse & Böhme, 2024), PrimeVul (Ding et al., 2024), ICVul (Lu et al., 2025). The last four rows show the combined datasets used in this work.

| Year | Dataset | Size | | | CVEs | Languages | Included | | | |
|------|---------|------|------|-------|------|-----------|--------|--------|--------|--------|
| | | Repos | VFCs | ¬VFCs | | | $DS_1$ | $DS_2$ | $DS_3$ | $DS_4$ |
| 2017 | Secbench | 238 | 676 | - | 98 | 13 PLs | ✓ | ✓ | ✓ | ✓ |
| 2019 | Project KB | 205 | 1282 | - | 624 | Java | ✗ | ✗ | ✗ | ✓ |
| | Devign | 2 | 10 894 | 14 978 | - | C/C++ | ✓ | ✓ | ✓ | ✓ |
| 2020 | Big-Vul | 348 | 4432 | - | 3754 | C/C++ | ✗ | ✓ | ✓ | ✓ |
| 2021 | CVEfixes | 4188 | 12 107 | - | 11 873 | 27 PLs | ✗ | ✓ | ✓ | ✓ |
| | SPI | 2 | 10 894 | 14 979 | 1045 | C/C++ | ✓ | ✓ | ✓ | ✓ |
| | PatchDB | 405 | 12 073 | 23 742 | 4076 | C/C++ | ✓ | ✓ | ✓ | ✓ |
| | TQRG21 | 1339 | 8057 | 110 161 | 5942 | 20 PLs | ✗ | ✓ | ✓ | ✓ |
| | CrossVul | 1675 | 5877 | - | 5131 | 48 PLs* | ✗ | ✓ | ✓ | ✓ |
| | CodeXGLUE | 4 | 10 894 | - | - | C/C++ | ✗ | ✗ | ✗ | ✗ |
| | 900Repo | 910 | 3765 | 6347 | 2460 | 20 PLs | ✗ | ✓ | ✓ | ✓ |
| 2022 | VCMatch | 10 | 1669 | - | 1669 | C/C++, J, PHP | ✗ | ✓ | ✓ | ✓ |
| | Tracer | 756 | 2898 | - | 2898 | 7+ PLs | ✓ | ✓ | ✓ | ✓ |
| | VUDENC | 784 | 1009 | - | - | Python | ✗ | ✗ | ✗ | ✗ |
| 2023 | DiverseVul | 797 | 7514 | - | - | C/C++ | ✗ | ✓ | ✓ | ✓ |
| | PySecDB | 351 | 1258 | 2791 | 479 | Python | ✗ | ✗ | ✗ | ✓ |
| | Sven | 269 | 559 | - | - | C/C++, Py | ✓ | ✓ | ✓ | ✓ |
| 2024 | MoreFixes | 7238 | 35 276 | - | 29 203 | 54 PLs | ✗ | ✗ | ✓ | ✓ |
| | MegaVul | 992 | 9019 | - | 8254 | C/C++, J | ✗ | ✓ | ✓ | ✓ |
| | VulnPatchPairs | 2 | 6352 | - | - | C/C++ | ✗ | ✗ | ✗ | ✗ |
| | PrimeVul | 755 | 6827 | - | 5369 | C/C++ | ✗ | ✗ | ✗ | ✗ |
| | RepoSPD | 364 | 18 124 | 31 394 | 2051 | C/C++ | ✓ | ✓ | ✓ | ✓ |
| 2025 | ICVul | 807 | 4327 | - | 4230 | C/C++ | ✗ | ✓ | ✓ | ✓ |
| | VFCDetective | 7967 | 61 667 | 133 954 | 9145 | | | | | |
| | $DS_1$ | 318 | 18 119 | 36 488 | 2507 | C/C++ | | | | |
| | $DS_2$ | 388 | 21 027 | 38 412 | 4051 | C/C++ | | | | |
| | $DS_3$ | 388 | 21 027 | 38 412 | 9145 | C/C++ | | | | |
| | $DS_4$ | 7967 | 61 667 | 133 954 | 9145 | ALL | | | | |

* Unique file extensions.
† Defect detection set is identical to Devign Zhou et al. (2019), CodeXGLUE contains several other tasks
‡ We were unable to gain access to the dataset
§ Advanced pattern matching using Prospector Sabetta et al. (2024)

Table 6: Inference benchmarking results on H200 NVL GPU. All models evaluated on 1000 sequential samples (no batching) after warmup. Time reported as mean ± std in milliseconds.

| Model | Block Size | Time (ms) | Peak Mem (MB) | Throughput (it/s) |
|---|---|---|---|---|
| CodeBERT | 512 | 6.23 ± 2.94 | 546 | 160 |
| CodeBERT C++ | 512 | 6.19 ± 3.00 | 546 | 161 |
| UniXcoder$_{512}$ | 512 | 6.18 ± 3.18 | 550 | 162 |
| UniXcoder$_{1024}$ | 1024 | 10.40 ± 2.96 | 630 | 96 |
| CommitBart | 1024 | 16.06 ± 0.08 | 1071 | 62 |
| CodeT5 Large | 512 | 34.95 ± 0.07 | 3265 | 29 |
| StarCoder2-15B | 2048 | 184.58 ± 35.45 | 30 183 | 5.4 |
| Qwen3-Coder 40B | 4096 | 2585.69 ± 70.12 | 61 854 | 0.4 |

### A.4 TOKEN SIZE STATISTICS

Table 8 provides detailed token size statistics for $\mathcal{D}_1$ across different input representations. The analysis uses the CodeBERT tokenizer and shows the distribution of samples across standard transformer context windows.

Table 8: Token size statistics for $\mathcal{D}_1$ using CodeBERT tokenizer. Values show count and percentage of samples within each token range.

| Level | Mean | Median | 0–63 | 64–127 | 128–255 | 256–511 | 512–1023 | 1024+ |
|---|---|---|---|---|---|---|---|---|
| Message | 133.0 | 96 | 33.6% | 31.1% | 25.7% | 7.4% | 1.5% | 0.7% |
| Commit | 607.2 | 249 | 4.4% | 15.3% | 31.4% | 23.9% | 13.4% | 9.9% |
| diff | 475.2 | 122 | 29.7% | 21.4% | 17.9% | 13.0% | 8.9% | 7.5% |
| diff$_{cf}$ | 792.7 | 170 | 13.2% | 25.8% | 23.9% | 14.6% | 9.2% | 9.9% |
| diff$_{df1}$ | 953.5 | 247 | 8.2% | 18.4% | 24.5% | 19.6% | 12.7% | 12.6% |
| diff$_{df2}$ | 1366.5 | 450 | 6.9% | 11.1% | 15.9% | 19.8% | 18.7% | 21.2% |

### A.5 COMMENT REMOVAL

To prevent comments from introducing spurious correlations or polluting our training samples, we implement a comment-aware preprocessing step that operates directly on git diff patches. While complete comment removal at the diff level presents unique challenges, particularly for multi-line comments that may span across different diff hunks or be partially included in the patch context, we adopt a pragmatic line-by-line approach. Our method processes each modified line (those prefixed with '+' or '-') or statement individually, preserving string literals while removing both single-line (//) and multi-line (/* */) comments. Specifically, we traverse each line character-by-character, tracking whether we are inside a string literal (respecting escape sequences) to avoid incorrectly removing comment-like syntax within strings (e.g., URLs containing '//'). This approach successfully handles the majority of comment patterns encountered in practice, including comments appearing after code statements, while maintaining the structural integrity of the diff format. Although this method has known limitations with multi-line comments spanning multiple diff lines, we find it provides an effective balance between implementation complexity and comment removal accuracy, ensuring that our transformer models focus primarily on code changes rather than documentation artifacts that may correlate with but not causally indicate VFCs.

### A.6 TRAINING STABILITY

Training the same classifier on the same train/ evaluation split for different seeds reveals very stable training results (see Figure 5). This stability of evaluation results has been observed during the development of this work and has motivated us to reduce the environmental impact of large scale training by mitigating statistical variations for all runs. To investigate the impact of the chosen split for repository-based splitting we split the data with three different seeds and evaluated the performance on different test sets. While this caused slightly higher variations, the training and performance are

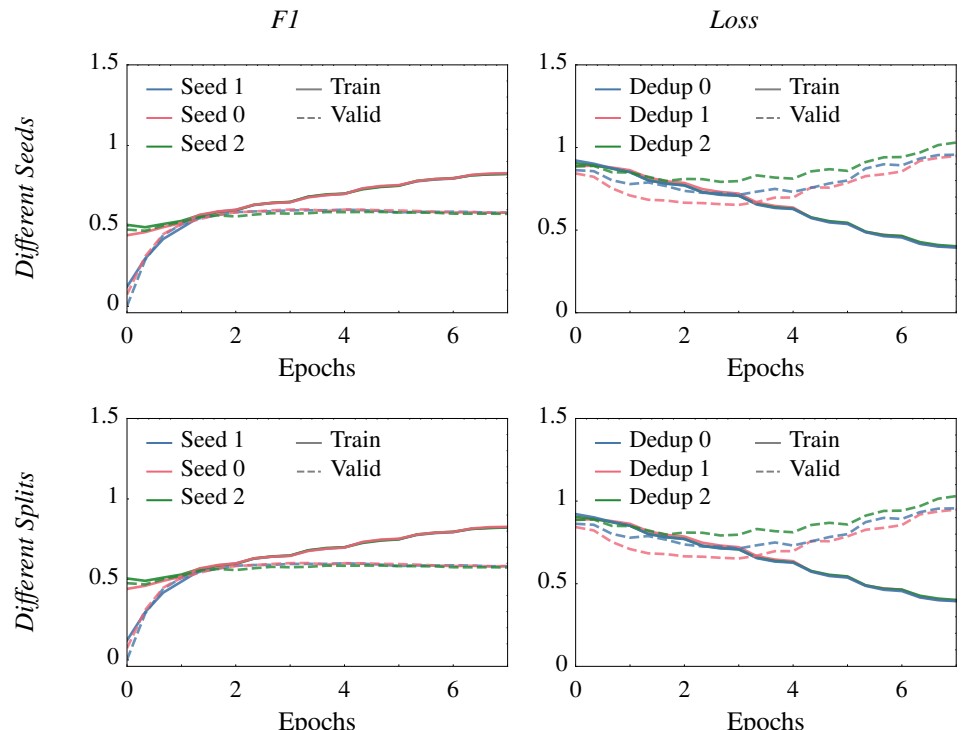

Figure 5: Training dynamic observed when training CodeBERT on `diffs` ($\mathcal{D}_1$) using different seeds on the same (repository) split and for different splits of $\mathcal{D}_1$.

still very aligned throughout all splits. To ensure this does not influence the development, we ran the second experiment after the evaluation was complete.

## A.7 TRAINING DYNAMICS

**Baseline Representations** Figure 6 illustrates the training loss and evaluation F1-Score for the three different input representations without context enrichment using a repository-based split. Both the metrics converge aligned for the model trained on commits and commit messages respectively. However, when only the diff is used as input, the training loss converges slower, and the evaluation F1-Score stagnates faster at a lower value. This underlines the performance on the test-set that was already observed in Table 2 while also demonstrating stable training. In general, the loss divergence after 2 epochs suggest that models start to memorize training-specific patterns early in the training.

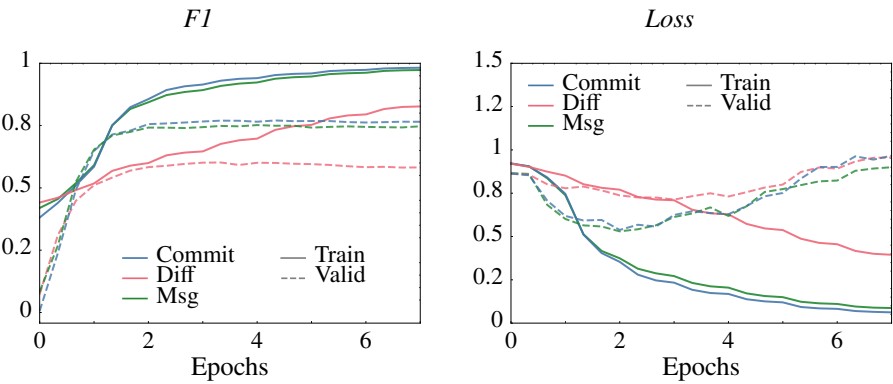

Figure 6: Training loss and evaluation F1-score across for the three different representations using a repository-based split.

**Splitting Strategy** This section provides additional insights into the training dynamics observed during our experiments on different data splitting strategies. The training loss and evaluation F1-Score for CodeBERT (Figure 7) highlight consistent convergence of the training loss with a divergent evaluation F1-Score across all splitting strategies. It is apparent that for repository-based splits, the training loss converges slower. Additionally, the evaluation F1-score slowly converges towards 1 while the evaluation F1-score stagnates quickly for all splits. However, when the models are trained exclusively on `diffs` a notably lower evaluation F1-Score can be observed.

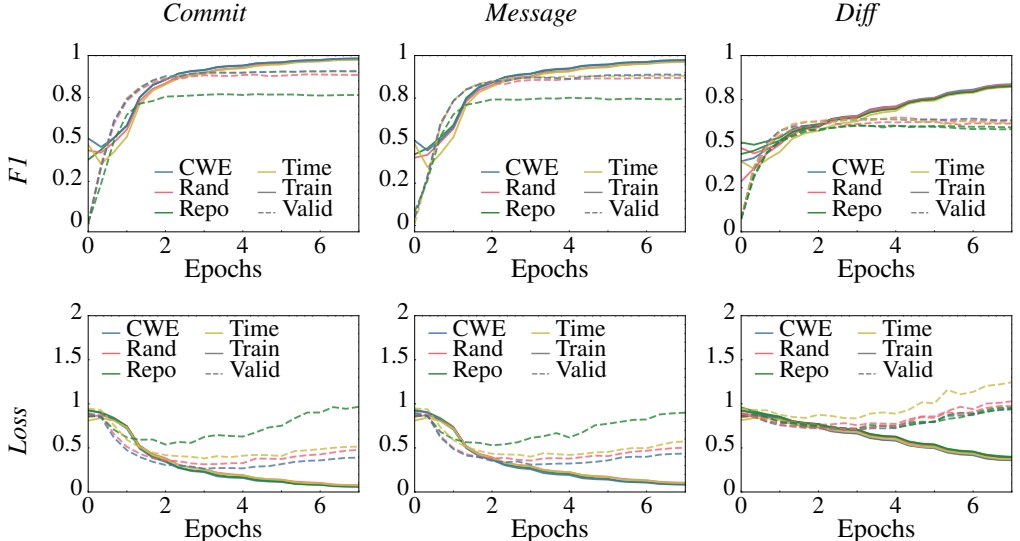

Figure 7: Training loss and evaluation F1-score evolution across three splitting strategies for Code-BERT.

**Loss Evolution** To gain a better understanding of the training dynamics observed during our context enrichment experiments, we provide the training loss evolution for all model configurations in Figure 8. Losses are grouped by representation (rows) and different model comparisons (columns) to visualize difference in training behaviour. While training dynamics are similar, several trends can be observed. Training and evaluation loss universally diverge between epoch 2 and 4 while for larger models higher overfitting can be observed causing larger loss divergence. The consistency of these dynamics across model architectures reinforces our finding that current transformer architectures struggle to effectively leverage semantic context for SPD.

**Label Quality vs. Quantity** To investigate the trade-off between dataset size and label quality, we evaluate models trained all four datasets proposed in Section 3. We run two sets of experiments, first we train and evaluate on splits of the same dataset and then we additionally draw evaluation and test sets from $\mathcal{D}_1$ following repository-based splitting. We train both CodeBERT and CodeT5 Large and show the results in Table 9. Testing on $\mathcal{D}_1$ provides insight into whether models trained on noisier but larger datasets can generalize to high-quality labels, while testing on the training distribution ($\mathcal{D}_2/\mathcal{D}_3$) reveals whether the additional samples compensate for label noise. Figure 9 presents the accompanying training dynamics. While the training dynamics do not reveal notable deviations, the model trained on $\mathcal{D}_2$ fails catastrophically on the test set, achieving a precision of 0.16 and an F1-score of 0.25. However, when the model is cross-trained on $\mathcal{D}_1$, the performance recovers. In general, this trend can be observed, that cross-training on $\mathcal{D}_1$ achieves higher results than the evaluation on the respective dataset, potentially due to the increased label noise in the respective datasets.

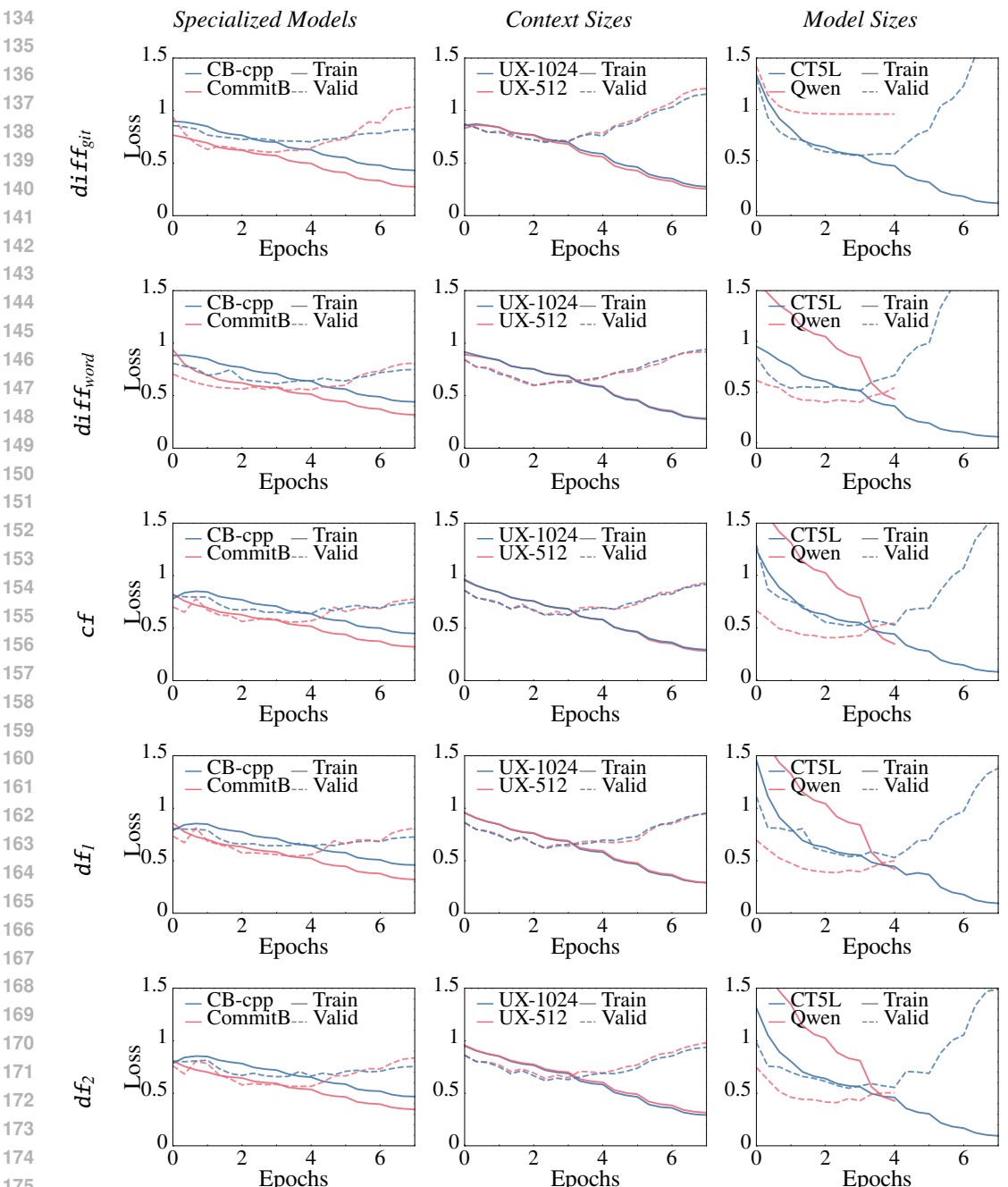

Figure 8: Training loss evolution for context enrichment experiments.

## A.8 PROMPT-BASED CLASSIFICATION PROMPT

For our prompt-based classification experiments using chain-of-thought (CoT) reasoning with git-formatted diffs, we employ the following prompt template. The system message establishes the security expert persona and provides instructions for analyzing code changes, while the user message contains the diff to be classified.

Table 9: Impact of training data quality vs. quantity on VFC detection performance. Models trained on advisory-based labels (DS2, DS3) vs. manually reviewed labels (DS1).

| Train | Test | CodeBERT | | CodeT5 Large | |
|---|---|---|---|---|---|
| | | F1 ↑ | PD-S ↓ | F1 ↑ | PD-S ↓ |
| $\mathcal{D}_2$ | $\mathcal{D}_2$ | 0.25 | 0.96 | 0.30 | 0.98 |
| | $\mathcal{D}_1$ | 0.56 | 0.99 | 0.56 | 0.99 |
| $\mathcal{D}_3$ | $\mathcal{D}_3$ | 0.51 | 0.96 | 0.52 | 0.97 |
| | $\mathcal{D}_1$ | 0.61 | 0.86 | 0.62 | 0.86 |
| $\mathcal{D}_4$ | $\mathcal{D}_4$ | 0.59 | 0.94 | 0.62 | 0.98 |
| | $\mathcal{D}_1$ | 0.63 | 0.86 | 0.62 | 0.89 |

$\mathcal{D}_1$: Manually reviewed labels, $\mathcal{D}_2$: Advisory-based labels (C/C++), $\mathcal{D}_3$: Combined labels (C/C++). PD-S: Patch Detection Score (FNR @ FPR ≤ 0.5%).

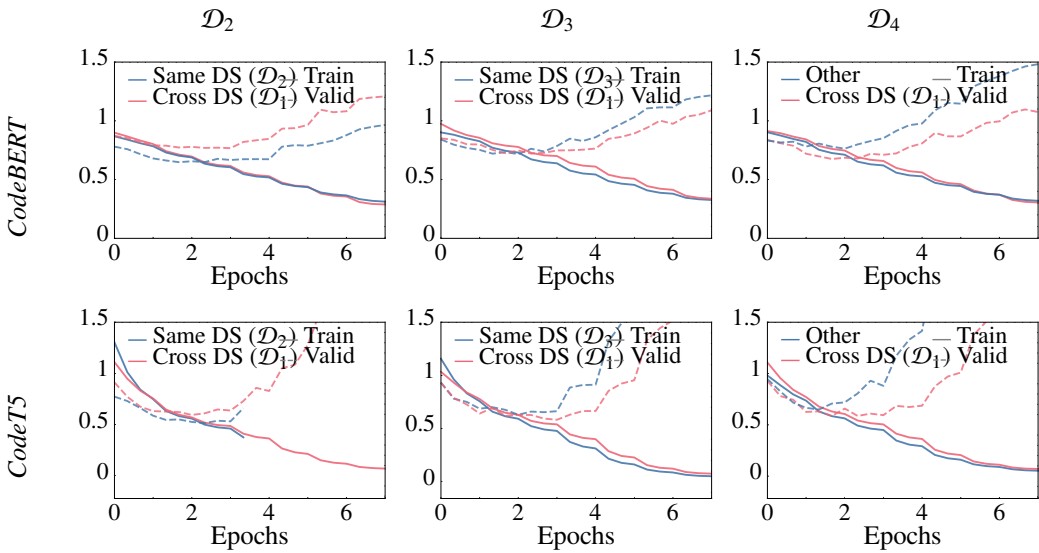

Figure 9: Training loss evolution across different datasets ($\mathcal{D}_2$, $\mathcal{D}_3$, $\mathcal{D}_4$) for CodeBERT and CodeT5 using a repository-based split. For each dataset, the evaluation split is once drawn from the same dataset and once from $\mathcal{D}_1$.

---

**System Message**

You are a software security expert with extensive experience in vulnerability assessment.
You are given a code diff from a commit of a software project. Based on the changes in the diff, you need to determine whether the commit is a vulnerability fixing commit (VFC) or a benign change that does not clearly indicate a vulnerability (BENIGN).

The diff is formatted in standard git diff format:
- Lines removed from the old version are prefixed with -
- Lines added to the new version are prefixed with +
- All other lines are context showing surrounding code

Think step-by-step:
1.  Analyze what changed in the diff
2.  Identify any security-related patterns (buffer overflows, injection flaws, authentication bypasses, memory safety issues, input validation, cryptographic weaknesses, etc.)
3.  Determine if the changes fix a vulnerability or are benign changes
4.  Provide your final classification

You MUST make sure only actual VFCs are labeled as such to minimize false positives.

Format your response as:
REASONING: [Your step-by-step analysis]
CLASSIFICATION: [VFC or BENIGN]

**User Message**

```
Diff:
{diff}

Provide your analysis:
```

