# OpenReview forum: "A Comprehensive Evaluation of Code Language Models for Security Patch Detection"
_ICLR.cc/2026/Conference — Submitted to ICLR 2026_

### Official Review · Reviewer_xtAD · 2025-10-20

**Soundness:** 3
**Presentation:** 3
**Contribution:** 1
**Rating:** 4
**Confidence:** 5

**Summary:**

This paper presents a comprehensive evaluation of transformer-based VFC detection through a unified framework consolidating 20 fragmented datasets spanning more than 180,000 commits.
Its analysis reveals that high performance metrics mask fundamental limitations.

**Strengths:**

- This paper focuses on addressing an important AI4SE question
- This paper conducts a comprehensive experiments on VFC detection.

**Weaknesses:**

Actually, I really appreciate the authors' efforts in conducting an evaluation on existing VFC detection datasets and approaches.

However, the main limitation is the insufficient study on related work [VulFixMiner, Sun et. al., Steenhoek et. al., ColeFunda, LLM4VFD], thus "re-finding" some existing findings.
- "Repository-based evaluation vs commonly-used temporal splits" please refer to [Steenhoek et. al.] Figure 5. Note that deep-learning-based vunlerability detection is highly-correlated with deep-learning-based vulnerability patch detection, this finding in this paper is not novel enough. Addtionally, VulFixMiner (ASE'21) also points out this issue and splits their training/validation/testing set in a repo-level.
- "the reliance on textual shortcuts" please refer to [Steenhoek et. al.]. That paper (also focusing on vulnerability patch identification) has already pointed out this fact in Table IV, i.e., commit message only performs better than code only, and including all codes is not a proper choice.

"achieves 33× speedup over existing approaches". This speedup mainly comes from the poor efficiency of joern, which generates much unnecessary information.

Model Selection. LLMs used in evaluation is a little outdated (the latest is StarCoder). Please consider more recent LLMs, which has shown substantially better performance in code-related tasks, e.g., Qwen3-Coder (the authors can also choose models with similar parameters)

#### Please add the following reference paper
[VulFixMiner] Zhou J, Pacheco M, Wan Z, et al. Finding a needle in a haystack: Automated mining of silent vulnerability fixes[C]//2021 36th IEEE/ACM International Conference on Automated Software Engineering (ASE). IEEE, 2021: 705-716.

[Sun et. al.] Jiamou Sun, Zhenchang Xing, Qinghua Lu, Xiwei Xu, Liming Zhu, Thong Hoang, and Dehai Zhao. 2023. Silent Vulnerable Dependency Alert Prediction with Vulnerability Key Aspect Explanation. In Proceedings of the 45th International Conference on Software Engineering (ICSE '23). IEEE Press, 970–982. https://doi.org/10.1109/ICSE48619.2023.00089

[Steenhoek et. al.] Steenhoek B, Rahman M M, Jiles R, et al. An empirical study of deep learning models for vulnerability detection[C]//2023 IEEE/ACM 45th International Conference on Software Engineering (ICSE). IEEE, 2023: 2237-2248.

[ColeFunda] Zhou J, Pacheco M, Chen J, et al. Colefunda: Explainable silent vulnerability fix identification[C]//2023 IEEE/ACM 45th International Conference on Software Engineering (ICSE). IEEE, 2023: 2565-2577.

[LLM4VFD] Xu Yang, Wenhan Zhu, Michael Pacheco, Jiayuan Zhou, Shaowei Wang, Xing Hu, and Kui Liu. 2025. Code Change Intention, Development Artifact, and History Vulnerability: Putting Them Together for Vulnerability Fix Detection by LLM. Proc. ACM Softw. Eng. 2, FSE, Article FSE023 (July 2025), 22 pages. https://doi.org/10.1145/3715738

[CompVPD] Chen T, Li L, Qian T, et al. CompVPD: Iteratively Identifying Vulnerability Patches Based on Human Validation Results with a Precise Context[J]. arXiv preprint arXiv:2310.02530, 2023.

**Questions:**

1. Please refer to the preceding weakness.
2. Note that existing LLMs are not trained with Gum-Tree's git diff formats, what about using raw git diff for LLM to comprehend?
3. What is the reference link of "[anonymous submission]" in the Section Data Availability?

---

> ### Author Response · Authors · 2025-11-20
>
> Dear Reviewer,
>
> we sincerely thank you for your review and especially the highlighting of important work that we missed. We have already included all mentioned works into the paper and rephrased the findings to align with and highlight that our respective findings strengthen the existing observations. We have generated a latexdiff pdf of the these changes specifically that we could share.
>
> Regarding the diff format, we initially had the same question and ran all experiments on both representations. Initial experiments showed that using the slightly more compact notation we use in the paper procudes slightly better results, hence we started focusing on this format. However, we will run the experiments from Table 3 again using the standard diff format as an ablation study that we will add to the appendix. We will provide the numbers once the training for this is complete.
>
> Thank you also for your suggestion of using a newer model, we agree that this makes a lot of sense and are currently running the experiments from Table 3 using Qwen3-Coder 40B. However this also poses some design decision we would like to discuss. At the moment the training is running with the same setup we used for the other models except with twice the context size. This means we attach a classification head to the internal state of the model and provide the sample without the (expected) prompt template to the model. We have decided to use this approach for comparability but are unsure if we should rather or also use the expected prompt template or attempt to fine tune the model in the generative setting. However for the last point the number of samples could be too limited to get stable generative results. We will also use this model without finetuning as a generative prompting based evaluation (see Reviewer DdsB).

---

> > ### Comment · Reviewer_xtAD · 2025-11-20
> >
> > Thanks for the authors efforts.
> >
> > As for the discussion question (in the third paragraph), it is proper to choose each experiment setting if it is clearly explained in the revised paper/rebuttal content.
> >
> > It seems that the PDF file has not been updated. Looking forward to the authors' updated version.

---

> > > ### Author Response · Authors · 2025-11-20
> > >
> > > We have now updated the pdf with the changes made to include the mentioned works and associated findings. We will additionally discuss the model agreement rates observed by [Steenhoek et. al.] after we performed qualitative failure case analysis as discussed with Reviewer DdsB and Zcem.
> > >
> > > In summary we have made the following changes:
> > > - We modified the abstract, introduction and conclusion to highlight and align with existing findings.
> > > - We added the respective works to the related work section.
> > > - In 5.2 Experimental results we added short discussions for both the commit message and cross-project results.
> > >
> > > Training for Qwen is still underway, we will update the paper with the results as soon as the training finishes.

---

### Official Review · Reviewer_Zcem · 2025-10-26

**Soundness:** 3
**Presentation:** 2
**Contribution:** 2
**Rating:** 4
**Confidence:** 3

**Summary:**

This paper presents a comprehensive and critical evaluation of transformer-based code language models (LMs) for the task of SPD. The authors construct a unified framework that consolidates 20 fragmented datasets, encompassing over 180,000 commits, to enable systematic comparison. The core finding is that the high performance reported in prior work is largely illusory, driven by models exploiting textual shortcuts in commit messages rather than learning the semantic patterns of security-relevant code changes. When restricted to code diffs alone, model performance drops precipitously (F1 ~0.6). The study further demonstrates that neither scaling model size (from 125M to 15.5B parameters) nor augmenting diffs with intra-procedural semantic context yields meaningful improvements. Evaluations using a rigorous repository-based split reveal significant performance drops (10-11%), indicating poor generalization and over-reliance on project-specific patterns.

**Strengths:**

•	Comprehensive Scope: The evaluation is unparalleled in its breadth across models, data, and experimental conditions.
	•	Rigorous and Realistic Evaluation: The use of repository-based splits and the PD-S metric provides a much more realistic and trustworthy assessment of model capabilities.
	•	Actionable Insights: The paper moves beyond simply reporting poor performance to diagnose the root causes: shortcut learning, lack of generalization, and architectural limitations.
	•	High Practical Utility: The release of the unified framework and preprocessing tools ensures high impact and promotes reproducibility and future research.

**Weaknesses:**

•	Label Quality Uncertainty: While acknowledged, the impact of noisy labels (e.g., in D3/D4 from ML/tool-based sources) isn't quantified via sensitivity analysis—could confound low code-only performance.
	•	Limited Scope in Evaluation: Focuses heavily on C/C++ (D1-D3); D4's multi-language inclusion is promising but underexplored (e.g., no cross-lang transfer results). Evaluation omits runtime metrics (e.g., inference speed) despite large models like StarCoder.
	•	Context Enrichment Limitations: Targets intra-procedural only; inter-procedural/cross-file dependencies (key in real patches) are unaddressed. Word-level diffs reduce size but may lose nuance in complex changes.
	•	Missing Baselines: Compares to graph methods (Wang et al., 2023) but not recent non-transformer approaches (e.g., dynamic analysis in Luo et al., 2024) or fine-tuned vision models on code graphs.
	•	Quantitative Gaps: Inter-model agreement is mentioned qualitatively; a table of pairwise agreements could strengthen claims of convergent (non-complementary) learning.
	•	Limited Exploration of the "Why": While the paper excellently documents what is not working (models don't understand code semantics), it offers less insight into why transformer-based architectures fail here. Is it a data-hunger issue, a fundamental limitation of the architecture for representing code changes, or a need for different pre-training objectives?
	•	Context Enrichment's Limited Scope: The context enrichment is strictly intra-procedural. The negative result, while valuable, leaves open the question of whether more costly but semantically richer inter-procedural or repository-level context (as in RepoCPG) could help.
	•	No Exploration of Very Recent LLMs: The model suite, while diverse, does not include the very latest generation of large language models specifically designed for code (e.g., CodeLlama, DeepSeek-Coder). Their performance might differ, though the fundamental limitations identified would likely persist.

**Questions:**

•	How sensitive are results to label noise? Did you perform robustness checks (e.g., dropping subsets with low manual verification) or estimate error rates via sampling?
	•	For context enrichment, what are the precision/recall of slicing at depths 1 vs. 2, and how does it handle non-C languages (e.g., Python's dynamic typing)?
	•	In repository-based splits, how did you select hold-out repos (e.g., size-matched to train?)? Any results on zero-shot transfer to unseen vulnerability classes?
	•	Given message reliance, have you analyzed specific textual patterns (e.g., keywords like "fix CVE") via interpretability tools, and do they correlate with false positives?
	•	The paper mentions releasing the framework—will it include pre-computed enriched diffs for all datasets, and what's the planned timeline/venue (e.g., GitHub)?
	•	Given the high model agreement rates, what do you hypothesize are the specific, but insufficient, "patterns" in the code diffs that all models are converging on? Have you analyzed the false positives/negatives to identify these common patterns?
	•	The results suggest a fundamental ceiling for current architectures. What, in your view, are the most promising alternative architectural approaches for this task? For example, should the community invest more in graph-based models, or models that explicitly reason over program semantics (e.g., using symbolic execution)?
	•	Beyond intra-procedural context, what other types of information or context do you believe are crucial for accurately identifying security patches from code alone? For instance, is cross-file data-flow, commit history, or a formal specification of the vulnerability necessary?

---

> ### Author Response · Authors · 2025-11-20
>
> Dear Reviewer,
>
> we sincerely thank you for your review. Below we attempt to address your questions and comments.
>
> **Question 1: Label Noise**
> We have not quantified label noise but decided to only include datasets in DS1 (Table 2) where the original authors have performed some level of manual validation. As we did not enrich the datasets with novel samples the original noise estimations are still valid. Additionally we will manually review samples that have high model agreement rates (Appendix, Figure 9) but are misclassified and samples that have low model agreement. We hope that this provides inside into mislabelling and failure cases.
>
> **Question 2: Context Enrichement**
> We have implemented the context enrichment only for C/C++ as this represent the largest set of commits. Regarding the precision/ recall, we are not sure how to answer this questions. From a technical perspective the algorithm walks along def-use chains, and should have a very low error rate. However from the semantic perspective, i.e. how much of the relevant context for the problem at hand is actually fetched, seems hard to quantify. We could perform manual analyis of some samples to attempt to quantify this empirically. But we suspect that in general, both metrics fall short as the data-flow seeded in changed statements is in itself only a proxy for the actual target. Especially in cases when the actually vulnerable statement (i.e. the sink) is not part of the set of changed statements this approach will fall short. Do you have any suggestion on how to effectively quantify this?
>
> **Question 3: Repository-based splits**
> We exlude repositories that are larger than the target test and evaluation set (i.e. linux) and then randomly sample repositories until the target size is reached.
> We did not train on specific vulnerability classes as this would require reducing the training set (removing all commits without vulnerability type information). Additionally the number of samples per class varies strongly, but we could run that experiment.
>
> **Question 4: Textual patterns**
> We have evaluated token importance for the classification result and have anecdotally seen strong token importance whenever a security relevant keyword is present in the message or even in the code. However we did not find significant correlations between single keywords and the label. We did not specifically look at false positives.
>
> **Question 5: Framework**
> Unfortunately we cannot not provide precomputed (enriched) commits due to licencing contraints in the source repositories as well as access requirements by the datasets we aggregated. However the framework provides all tooling required to reconstruct the commits as well as the enrichement. We could potentially provide precomuted samples in cases where the preconditions allow it (a smaller set).
>
> **Question 6: Code Patterns**
> We have not analysed that but as also requested by other reviewers we will perform an additonal qualitative failure case analysis based on the model agreement rates (Appendix, Figure 9). We will analyse samples that have high model agreement but are missclassified and samples that have low model agreement. This could offer insights into both mislabelling and common failure cases.
>
> **Question 6/7: Outlook**
> While this is undoubtably a very difficult question to answer, we suspect that finding representations of code that better capture the inherent properties or semantics of the underlying problem (be that VD or SPD) would undoubtably help. The type of representation could then inform the model choice. However, with the recent advancement made on ReAct style agent systems that are capable of dynamically exploring the program space while keeping relatively large contexts could allow for different ways of representing relationships inside code that allow for efficient model traversal. Yet fundamentally, defining representations that efficiently capture a robust view of problem would likely benefit all models.
>
> We will consider this a bit more and add a small discussion to the end of the paper.
>
> **Additional Comments:**
> - **Inter-procedural context** We have initially built an additional representation that also captures upstream functions from the calling context to provide more context. We evaluated this on Anthropic's Opus (due to context size). As the results were not promising we did not include them but we will now rerun these experiments and add Qwen3-Coder as an open-source alternative and report the findings in the paper.
> - We will also add inference speed and memory usage as **runtime metrics** to the paper.
> - For **inter-model** agreement (Appendix, Figure 9) you mentioned a Table of pairwise agreements, could you further elaborate what this table should look like? We would appreciate the input here and are happy to extend the analysis.
> - As mentioned with Reviewer xtAD, we are currently training Qwen3-Coder as a recent 40B model to extend the evaluation to more modern models.

---

### Official Review · Reviewer_iruk · 2025-10-29

**Soundness:** 4
**Presentation:** 3
**Contribution:** 3
**Rating:** 6
**Confidence:** 3

**Summary:**

Core Strengths

Unified Dataset Framework: Addressing Fragmentation in the FieldExisting research on Vulnerability-Fixing Commit (VFC) detection is limited by fragmented datasets—different studies use small-scale datasets with independent annotations and inconsistent formats (e.g., PatchDB, Devign), making horizontal comparisons difficult. This paper is the first to integrate 20 cross-language, cross-vulnerability-type datasets (covering over 180,000 commits). Through standardized parsing, deduplication (three strategies: hash matching, semantic matching, and UniXcoder embedding-based deduplication), and filtering (supporting filtering by language and vulnerability type), it constructs a reusable unified framework. This contribution fills a gap in the field, provides a "fair comparison benchmark" for subsequent research, and can significantly reduce the verification cost of new methods after open-sourcing.

Revealing Core Limitations of Code LMs: Reliance on Textual Shortcuts Over Semantic UnderstandingPrevious studies (e.g., Tang et al. 2023, Wang et al. 2021b) only reported high F1 scores (~0.9) for "code + commit message" but failed to decompose the contributions of the two components. Through ablation experiments, this paper finds that:
When only code is used, the F1 score drops sharply to 0.6;
When only commit messages are used, the F1 score remains at 0.88 (close to the performance with full information).
This finding subverts the perception that "code LMs already possess security semantic understanding capabilities" and clearly indicates that existing models rely on textual keywords such as "security" and "fix" rather than truly understanding the security significance of code modifications, pointing out a core direction for improvement in the field.

Rigorous Generalization Evaluation: Exposing Cross-Repository Transfer DefectsMost existing studies adopt "temporal splits" (dividing training/test sets by commit time), which mask the generalization defects of models. This paper is the first to introduce "repository-based splits" (training and test sets from different projects) and finds that performance decreases by 10-11% (e.g., CodeBERT’s F1 score drops from 0.89 to 0.79). Additionally, graph models like GraphSPD experience a 30% F1 drop in cross-repository scenarios. This result proves that models learn "project-specific patterns" (e.g., the code style of a specific repository) rather than universal security semantics, providing a key warning for model application in real-world scenarios (e.g., cross-project dependency detection).

Lightweight Context Enrichment: Efficiency BreakthroughTo address the high preprocessing latency of existing graph models (e.g., GraphSPD, 57.36 seconds per sample), this paper proposes an "intra-procedural context enrichment method" based on data flow and control flow, achieving a 33x speedup (1.72 seconds per sample). Although it does not improve performance, it provides an efficient foundation for subsequent context optimization (e.g., reducing computational costs when extending to inter-procedural context).

**Strengths:**

Application Strengths

1. Open-Source Framework Empowers Industry and Academia

The paper open-sources the unified framework and evaluation suite, allowing researchers to quickly access 20 datasets and reproduce experimental results, significantly lowering the entry barrier for VFC detection research. For industry, the framework can be directly used for benchmark testing of internal vulnerability detection systems (e.g., evaluating the performance of different models on enterprise private projects), accelerating technology deployment.

2. Providing Key Warnings for Industrial Deployment

The core findings of the paper (reliance on messages, poor cross-repository generalization) can guide industry in risk avoidance:
If commit messages are non-standard in a project (e.g., lacking keywords like "fix"), the performance of existing models will drop significantly, requiring supplementary manual reviews;
In cross-project dependency detection (e.g., supply chain security), direct use of existing models should be avoided, and targeted optimization of generalization capabilities is necessary.

3. Lightweight Methods Adapt to Resource-Constrained Scenarios

Although the 33x speedup context enrichment method does not improve performance, its efficiency has potential value in resource-constrained scenarios (e.g., edge devices, real-time code review tools). Subsequent optimizations can build on this to enhance semantic capture capabilities while balancing speed and performance.

**Weaknesses:**

1. Lack of core algorithmic innovation, only staying at the "evaluation" level without exploring solutions;
2. Insufficient coverage of industry-critical dimensions such as multilingual support, silent patches, and engineering metrics, weakening the application guiding value.

**Questions:**

1. Explore "Code Semantics-Oriented" Model Optimization Solutions

- Data Augmentation: Randomly shuffle or mask security keywords (e.g., "security," "fix") in commit messages to force models to focus on code modifications;
- Semantic Supervision: Introduce code semantic signals (e.g., AST structures, data flow dependencies, vulnerability type labels) and build multi-task training (e.g., "VFC detection + vulnerability type classification") to enhance the model’s understanding of security semantics;
- Contrastive Learning: Design positive and negative sample pairs (e.g., "real vulnerability fixes" vs. "semantically similar non-fix code modifications") to enable models to distinguish between "surface text" and "deep security semantics."
2. Extend Context Enrichment to the "Inter-Procedural" Dimension
The existing intra-procedural context cannot cover vulnerability fixes involving multi-function collaboration (e.g., "missing input validation in function A leads to buffer overflow in function B"). Suggestions include:
- Construct inter-procedural control flow/data flow graphs (e.g., including function call chains) to expand the context scope;
- Evaluate the detection improvement of inter-procedural context for "complex vulnerability fixes" (e.g., logical vulnerabilities, multi-function dependent vulnerabilities).
3. Supplement Specialized Evaluations for Multilingual and "Silent Patches"

- Build specialized datasets for languages such as Python and Java to evaluate model performance differences across languages (e.g., dynamic vs. static languages);
- Screen "silent patches" (without advisories or security keywords) from the dataset, separately evaluate the model’s F1 and PD-S on such samples, and quantify the model’s ability to address real pain points.
4. Incorporate Engineering Metrics for Industrial Evaluation
Supplement engineering metrics such as model inference time (time per sample), memory usage, and the trade-off curve between parameter quantity and performance to provide industry with a "performance-cost" selection basis (e.g., recommending the 125M-parameter CodeBERT instead of the 15.5B-parameter StarCoder, as they have similar performance but lower deployment costs).
5. Deepen Failure Case Analysis
Classify samples where the model fails to detect (e.g., missed detections, false detections) and analyze:
- Which vulnerability types (buffer overflow, SQL injection, logical vulnerabilities) the model performs worst on;
- Whether failed samples share the commonality of "complex code modification semantics but no message prompts," providing specific directions for subsequent optimization.

---

> ### Author Response · Authors · 2025-11-20
>
> Dear Reviewer,
>
> we sincerely thank you for your review. Below we attempt to answer your questions and indicate additional experiments we intend to include to answer them.
>
> **Question 1: Explore “Code Semantics-Oriented” Model Optimization Solutions**
> 1. **Data Augmentation** We have considered using masking but have decided that completely removing the commit message from the training (Table 2/3) is a stronger version of this with regard to the evaluation of the models code property understanding.
> 2. **Code Semantic Signals** The context enrichment essentially provides implicit data flow signals by adding relevant statements. We did not add explicit structures as the premise is to evaluate the capability of sequence based models that are trained on natural language. However considering hybrid models such as GraphCodeBERT could offer interesting insights.
> 3. **Vulnerability class prediction** We have considered this but did not train such a model as a) this would reduce the size of the dataset by removing commits for which no type information is available and b) many CWE classes are represented only by a small number of samples. We will add both a table breaking down the dataset by CWE or OWASP Top 10 and add an evaluation that looks at performances split in the evaluation set by vulnerability type. If you think it would be better to also train a model on a specific class despite the limited number of samples we could also add this as an additional experiment.
> 4. **Constrastive Learning** We have not performed contrastive learning to limit the scope of the study but we agree that this is an interesting area. One could attempt to specifically build pairs where the pretrained embeddings (cmp. Figure 3) from a standard model are close but the labels disagree.
>
> **Question 2: Extend Context Enrichment to the “Inter-Procedural” Dimension**
> 1. We have built an inter-procedural context generation step that fetches all functions that are above the target function in the call chain. We have also initially performed experiments with this on Anthropics Opus Model but did not see significant performance improvements. As discussed in the comment for Reviewer DdsB will rerun these experiments and add them to the paper along with Qwen3-Coder as an open-source alternative.
>
> **Question 3: Supplement Specialized Evaluations for Multilingual and “Silent Patches”**
> 1. **Multilingual** We have performed some experiments to see if model performance improves if various languages are seen during training (Appendix, Table 7). The framework we built supports extraction of datasets for all languages that are part of the overall collection however we have limited the evaluation to C/C++ as the context enrichement is also implemented for C/C++.
> 2. **Silent Patches** We have not performed that that analysis but we could add this as an ablation study. Hower in general, the models code-only performance should somewhat approximate this performance as it cannot rely on typical metadata?
>
> **Question 4: Incorporate Engineering Metrics for Industrial Evaluation**
> We have not yet measured this but we are happy to add additional information such as the inference time per sample for each model.
>
> **Question 5: Deepen Failure Case Analysis**
> Based on the model agreement rates (Appendix, Figure 9) we will analyse samples that have high model agreement but are missclassified and samples that have low model agreement. This could offer insights into both mislabelling and common failure cases.

---

### Official Review · Reviewer_DdsB · 2025-10-31

**Soundness:** 3
**Presentation:** 3
**Contribution:** 2
**Rating:** 4
**Confidence:** 4

**Summary:**

This paper evaluates code language models (125M–15B parameters) for Security Patch Detection (SPD) across 20 datasets, covering C/C++ projects such as Linux, FFmpeg, and Chromium. The authors introduce a unified VFC (vulnerability-fixing commit) framework and attempt to enhance model capability by injecting intra-procedural semantic context (AST, CFG, program slices). Results show that transformer-based code LMs primarily rely on commit messages rather than true code semantics, fail to generalize across repositories, and benefit minimally from syntactic/semantic augmentation.

**Strengths:**

- Timely and security-relevant problem.
- Thorough empirical effort across many datasets and sizes.
- Highlights real and dangerous LM failure modes in SPD.
- Repository-based split addresses realistic deployment drift.
1. **Insufficient baseline variety** — no symbolic/static tool integration.
2. **Overclaims** (“architectural limitations”) without hybrid evaluations.
3. **Under-explored analytical depth**: no CWE-class breakdown, patch difficulty taxonomy, or qualitative failure cases.
4. **No modern LLM prompting or RAG security baselines**.
5. Only intra-procedural semantics — **no inter-procedural reasoning** or software supply-chain context.

**Weaknesses:**

1. **Insufficient baseline variety** — no symbolic/static tool integration.
2. **Overclaims** (“architectural limitations”) without hybrid evaluations.
3. **Under-explored analytical depth**: no CWE-class breakdown, patch difficulty taxonomy, or qualitative failure cases.
4. **No modern LLM prompting or RAG security baselines**.
5. Only intra-procedural semantics — **no inter-procedural reasoning** or software supply-chain context.

**Questions:**

1. Did you quantify dataset noise or perform label disagreement analysis?
2. How do results vary by vulnerability type (e.g., buffer overflow vs logic)?
3. Did you try prompting GPT-4/Claude with chain-of-thought or tool use?
4. Why no comparison with symbolic/static analysis or hybrid approaches?

---

> ### Author Response · Authors · 2025-11-20
>
> Dear Reviewer,
>
> we sincerely thank you for your review. We would first like to attempt to answer your questions and then provide additional comments regarding your review.
>
> **Question 1: Did you quantify dataset noise or perform label disagreement analysis?**
> 1. **Label Noise** We did not attempt to quantify label noise explicitly as we believe this would require sampling sufficient samples from each underlying data source to gain statistically relevant results. However, the construction of our first dataset (_DS1_, Figure 2) relies on underlying sources that have each individually performed some level of manual validation of label noise. However we would like to perform some additional manual validation of samples that have a high model agreement rate (Appendix, Figure 9) and are misclassified as well as sampels with a low model agreement rate to see if these samples are often mislabeled. This evaluation will also be used to strengthen the analytical depth by attempting to identify patterns that cause misclassification.
> 2. **Label Disagreement Analysis** If this refers to the disagreement between the labels in the original data sources, then we have performed this analysis: Commits with conflicts that were excluded (_DS1_: 998, _DS2_: 4411, _DS3_: 5706, _DS4_: 5706). We will add these values to Chapter 3 where we introduce the respective datasets.
>
> **Question 2: How do results vary by vulnerability type (e.g., buffer overflow vs logic)?**
> We have not yet computed this but we will add an additional table to the appendix that breaks down the performance on the evaluation set for each sample where we are able to infer a CWE. We will provide these values as an additional comment in the next days.
>
> **Question 3: Did you try prompting GPT-4/Claude with chain-of-thought or tool use?**
> Yes we ran experiments on a smaller set of 850 samples using Anthropics Opus model with chain-of-thought. We tried using only the diff as well as the extended context that we have presented in the paper but have seen F1 scores between 0.58 and 0.60 across various context lengths. We have also included inter-procedural context by fetching the function code of all calling locations for the changed function as the context window is significantly larger for these models. However even then we did not see a noticable improvement in performance. Hence we initially left the results out of the paper but we are happy to rerun the experiments on a larger scale and include the results as part of the evaluation. We will also use Qwen3-Coder as an open-source model for comparison (which we are also including as part of the evaluated fine-tuned models, see Reviewer xtAD). Regarding **tool-use** we think that this is a very promising direction, however we have intentionally excluded that here as we feel this would require a significant number of ablation studies to justify design decisions and understand where the performance is gained.
>
> **Question 4: Why no comparison with symbolic/static analysis or hybrid approaches?**
> Dynamic approaches would generally require the software to be built/ buildable which is not the case here making it hard to use any tools that require execution of the target. Regarding static tools we were unsure which would be usable on this large variety of repositories that would also provide meaningful insights for SPD. If you would have any suggestions for tools that would be suitable we would be happy to attempt to run them on the data.
>
> **General Comments**
>
> We would like to address some of the weaknesses that you highlighted by:
> - (Re)running _chain-of-thought_ based prompting using both a commercial and open-source generative model. As these models allow for significantly more context size we will also include inter-procedural context. WE build this by including functions that call the target function(s).
> - Providing detailed breakdown of the models performances per CWE.
> - Analysing samples that have high model agreement but are missclassified and samples that have low model agreement. This could offer insights into both mislabelling and common failure cases.

---

### Author Response · Authors · 2025-12-03
**Updated version with requested experiments**

Dear AC,

We would like to briefly summarize the new experiments we ran based on reviewer feedback and how we incorporated new results and other comments into the revised manuscript.

In the first revised version, we addressed the very helpful review of Reviewer xtAD, who pointed out relevant missing related work. As described in more detail in our response to Reviewer xtAD, we have added all requested works and discussed how existing findings align with the results of this study.

To address the paper's weakness of missing recent language models and prompting-based evaluation raised by Reviewers DdsB, Zcem, and xtAD, we have extended the main experiments by also fine-tuning Qwen3-Coder 40B, which shows at best marginally higher performance compared to previous models (Table 3). Additionally, we have added a new experiment where we evaluate Qwen3-Coder 480B AWQ using two different prompting-based strategies (Table 4). Results show significantly lower performance than the fine-tuned models.

To address the under-explored analytical depth and failure case analysis raised by Reviewers DdsB and iruk, we performed a manual evaluation of the top 50 samples (sorted by average model confidence) where all eight models (from Table 3) agree but misclassify the sample. Especially for false positives, we were able to identify common patterns in the failure cases, described in the Failure Case Analysis section (Line 469).

To address missing runtime metrics raised by Reviewers iruk and Zcem, we have added detailed memory consumption and inference times to the appendix (Table 6) for all fine-tuned models.

Additionally, to address missing evaluations raised by Reviewers iruk and DdsB, we have added a new splitting strategy where the test set is built from commits associated with a CWE (Table 2) to evaluate approximate model performance on commits with high label accuracy from the real-world distribution. Initially, we wanted to further split this into groups of common CWEs but decided against this due to the limited number of samples per CWE.

Lastly, we intended to also add inter-procedural evaluations to the paper, as they were requested by Reviewers DdsB, iruk, and Zcem, but after another run of experiments, we decided against this. While we believe that this is a promising direction, we feel that a single approach to extending inter-procedural context that we implemented would not be sufficient. A comprehensive evaluation is beyond the scope of this work, which is why we have decided to defer this to future studies.

---

### Meta-Review · Area_Chair_QVcq · 2026-01-07

**Summary:**

This paper presents a large-scale and carefully executed evaluation of code language models for security patch (VFC/SPD) detection, unifying 20 datasets and systematically analyzing model behavior across input modalities, splits, and scales. Reviewers broadly agree that the empirical effort is substantial and that the paper clearly exposes important failure modes, particularly the heavy reliance on commit messages and poor cross-repository generalization. However, the contribution is primarily evaluative, and several reviewers raised concerns about limited novelty, overstatement of conclusions, and missing or underdeveloped comparisons that weaken the paper’s impact for ICLR.

**Reviewer Concerns:**

The authors made a genuine effort to address reviewer feedback. They added missing related work, included newer models (Qwen3-Coder 40B), conducted prompting-based evaluations with very large models, added runtime metrics, introduced a CWE-based split, and expanded failure case analysis. These additions improve completeness and presentation.

However, several core concerns remain only partially addressed:

The paper largely confirms and strengthens existing findings (e.g., message reliance, cross-repository degradation) rather than introducing new conceptual insights or methodologies.

Claims about “fundamental architectural limitations” remain insufficiently supported, as comparisons with hybrid, symbolic, or static-analysis-based approaches are still missing.

The analysis, while broader, remains diagnostic rather than explanatory: it documents what fails more convincingly than why, and does not translate findings into concrete methodological guidance.

The evaluation scope remains narrow in key dimensions (e.g., inter-procedural context, multilingual settings, silent patches), with several promising directions explicitly deferred.

**Reviewer Scores:**

Given the added experiments, some reviewers might slightly increase their confidence in the empirical thoroughness, but the fundamental concerns about contribution and novelty would likely persist. Overall scores would remain around the borderline-to-below-acceptance range.

---

### Decision · Program_Chairs · 2026-01-26

Reject